# Multi-objective Hyperparameter Optimization in the Age of Deep Learning

## Abstract

While Deep Learning (DL) experts often have prior knowledge about which hyperparameter settings yield strong performance, only few Hyperparameter Optimization (HPO) algorithms can leverage such prior knowledge and none incorporate priors over multiple objectives. As DL practitioners often need to optimize not just one but many objectives, this is a blind spot in the algorithmic landscape of HPO. To address this shortcoming, we introduce `PriMO`, the first HPO algorithm that can integrate multi-objective user beliefs. We show `PriMO` achieves state-of-the-art performance across 8 DL benchmarks in the multi-objective *and* single-objective setting, clearly positioning itself as the new go-to HPO algorithm for DL practitioners.

## 1 Introduction

Modern Deep Learning (DL) pipelines (Vaswani et al., 2017; Jumper et al., 2021; Brown et al., 2020) are highly sensitive to the choice of their hyperparameters, the manual tuning of which has become an increasingly time-consuming and costly task. Despite substantial advances in algorithms for Hyperparameter Optimization (HPO) (Bergstra et al., 2011; Li et al., 2017; Falkner et al., 2018; Mallik et al., 2023), many researchers continue to rely on manual tuning (Bouthillier & Varoquaux, 2020), which allows intuitive incorporation of domain expertise and prior beliefs about the best performing hyperparameter settings.

While HPO researchers have formulated desiderata for HPO algorithms that include incorporating such user beliefs, existing research has focused exclusively on single-objective optimization (Ramachandran et al., 2020; Souza et al., 2020; Hvarfner et al., 2022; Mallik et al., 2023). However, for DL, it is often necessary to optimize over several objectives, such as computational cost, training time, latency or fairness (Izquierdo et al., 2021; Schmucker et al., 2020; Salinas et al., 2021; Schneider et al., 2023). Thus, integrating prior knowledge into multi-objective optimization is a crucial research area that remains unexplored. We therefore adapt the desiderata for HPO algorithms for DL (Falkner et al., 2018; Mallik et al., 2023; Franceschi et al., 2025) as follows:

Table 1: Comparison of our algorithm `PriMO` to prominent categories of multi-objective algorithms with respect to the identified desiderata. The algorithmic categories include Evolutionary algorithms (EA, *e.g.* NSGA-II, SMS-EMOA), multi-objective multi-fidelity algorithms (MOMF, *e.g.*, MOASHA, MO-HyperBand, HyperBand with Random Weights), and multi-objective Bayesian optimization (MO-BO, *e.g.*, BO with random weights, BO with EHVI, ParEGO). A ✓ indicates that the method satisfies the criterion; a ✗ indicates it does not. (✓) denotes partial fulfillment or fulfillment with additional assumptions.

| Criterion | RS | EA | MOMF | MO-BO | PriMO |
|---|---|---|---|---|---|
| Utilize cheap approximations | ✗ | ✗ | ✓ | ✗ | ✓ |
| Integrate multi-objective expert priors | ✗ | ✗ | ✗ | ✗ | ✓ |
| Strong anytime performance | ✗ | ✗ | ✓ | ✗ | ✓ |
| Strong final performance | ✗ | (✓) | (✓) | ✓ | ✓ |

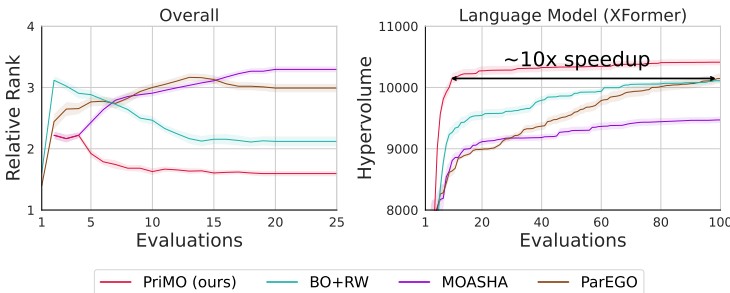

Figure 1: Comparison of `PriMO` and prominent multi-objective algorithms. [**Left**] Mean relative ranks across 8 DL benchmarks under all prior conditions averaged. [**Right**] Mean dominated Hypervolume for tuning the hyperparameters of a language model, demonstrating that `PriMO` can leverage a good prior to offer speedups of up to ~10x.

1. **Utilize cheap approximations**: Modern HPO algorithms must not only support optimization over multiple objectives, but should also be able to utilize cheap proxies of an objective function, if available, to speed up the optimization.

2. **Integrate multi-objective expert priors**: Expert prior knowledge of hyperparameters is often available for real-world DL tasks. A modern HPO algorithm must be able to utilize such beliefs over multiple objectives to speed up the optimization and be able to meaningfully recover from misleading prior information.

3. **Strong anytime performance**: Multi-objective HPO algorithms must be compute-efficient, *i.e.*, under limited budget, they must find candidates that significantly improve the dominated Hypervolume.

4. **Strong final performance**: The ultimate goal of HPO is to find the best performing configurations. As budgets grow larger, the algorithms should yield strong solutions.

Table 1 shows that existing HPO algorithms satisfy at most half of the criteria. To address this gap, we propose `PriMO`, which is the first HPO algorithm to incorporate expert knowledge over the optima of multiple objectives and also leverages cheap approximations of expensive objective functions. Our **main contributions** are as follows:

- We are the first to consider expert priors for multiple objectives (Section 2) and show that naively adapting existing algorithms is not a robust solution (Section 3).

- We introduce `PriMO`, a Bayesian optimization algorithm that integrates multi-objective expert priors in its acquisition function and exploits cheap proxy tasks in its initial design (Section 4). As such, `PriMO` is the first HPO algorithm to meet all the requirements of multi-objective HPO for practical DL (Table 1) and empirically yields up to 10x speedups over existing algorithms (Figure 1).

- We empirically demonstrate state-of-the-art performance of `PriMO` across a variety of DL benchmarks in the multi-objective *and* single-objective setting (Section 5.3). Furthermore, we show that `PriMO` is robust to different priors strengths (Section 5.4) and, in an ablation study, we verify that all components of `PriMO` are helpful and necessary (Section 5.6).

## 2 MULTI-OBJECTIVE HPO WITH EXPERT PRIORS AND CHEAP APPROXIMATIONS

To capture all the above desiderata, we propose the novel problem formulation of minimizing a vector-valued objective function $f$, while exploiting cheap approximations of its individual objectives and expert priors. For background on HPO for DL and the multi-objective case see Appendix B, and for a comparison to related problem formulations see Section 6.

**Introducing multi-objective expert priors** To extend expert priors over a single objective (Hvarfner et al., 2022) to the multi-objective setting, we consider a factorized prior as follows. For each objective $f_i$ of the vector-valued function $f$, prior beliefs $\pi_{f_i}(\lambda)$ represent a probability distribution over the location of the optimum of $f_i$. Specifically, the prior will have a high value in regions that the user believes have an optimum. Formally, we define

$$\pi_{f_i}(\lambda) = \mathbb{P}\left(f_i(\lambda) = \min_{\lambda' \in \Lambda} f_i(\lambda')\right), \tag{1}$$

yielding the compound prior $\Pi_f(\lambda) = \{\pi_{f_i}(\lambda)\}_{i=1}^n$, i.e., the set of prior beliefs over the optima of the individual functions that comprise $f$. For a discussion on our assumptions on priors see Section 7 and for details on the priors we consider in our experiments see Section 5.1. Further, we provide a discussion on the sources of prior knowledge in Appendix D.

**Integrating multi-objective expert priors and cheap approximations** To also leverage cheap approximations of the individual objectives, let $\hat{f}_i(\lambda, z)$ denote the low-fidelity proxy for $f_i$, where hyperparameters $\lambda$ are evaluated at the fidelity level $z$, where $f_i(\lambda) = \hat{f}_i(\lambda, z_{max})$. Therefore, our goal is to solve

$$\arg\min_{\lambda \in \Lambda} f(\lambda) = \arg\min_{\lambda \in \Lambda}\left(\hat{f}_1(\lambda, z_{max}), ..., \hat{f}_n(\lambda, z_{max})\right), \quad \text{guided by } \Pi_f(\lambda), \tag{2}$$

using inexpensive evaluations of $f$, while addressing the challenge that the priors may be misleading. Since the solution in multi-objective optimization is not a single optimum, but rather a Pareto front of trade-offs between objectives, our formulation seeks to guide the optimization process toward promising regions of this front.

## 3 Poor performance of the naive solution

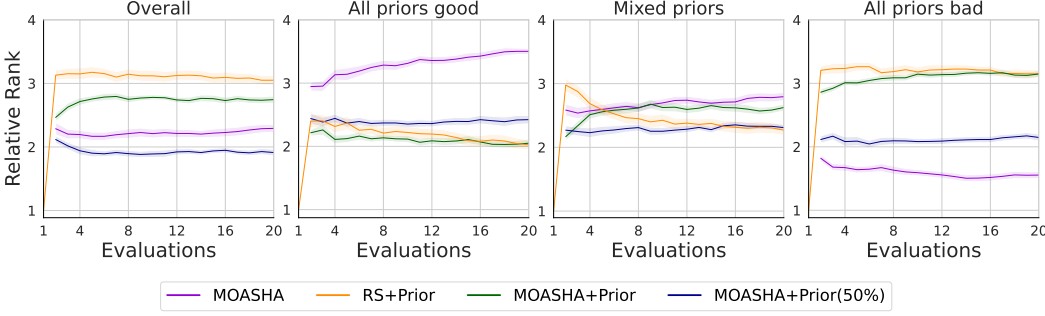

Figure 2: Mean relative ranks $\pm$ 1 standard error across benchmarks and seeds under various prior conditions for randomly sampling from the priors, MOASHA, and adaptations of it that utilize multi-objective expert priors. See Section 5.1 for details on the evaluation protocol.

In this section we study the effect of solving Equation 2 by naively adapting a multi-objective algorithm that can already utilize cheap approximations. We find that this naive solution does not perform robustly across different prior strengths (Figure 2), calling for a better-designed algorithm.

Specifically, we modify the random sampling in multi-objective asynchronous successive halving (MOASHA) (Schmucker et al., 2021), so that configurations are sampled from one of the priors $\pi_{f_i}$ chosen randomly at every iteration or 50% of iterations. In the presence of good prior knowledge, always sampling from the prior considerably outperforms standard MOASHA, but under misleading priors leads to drastically poor performance. 50% prior-sampling performs better than MOASHA overall, but is unable to effectively utilize good priors as well as 100% prior sampling or even prior-based random search. Therefore, we propose `PriMO`, to benefit from *good* priors while having the ability to recover from *bad* ones.

# 4 PRIMO: PRIOR INFORMED MULTI-OBJECTIVE OPTIMIZER

In this section, we introduce the first multi-objective HPO algorithm, `PriMO` (Algorithm 3), that leverages multi-objective user priors and fulfills all the desiderata of modern HPO.

We discuss how `PriMO`, a Bayesian Optimization algorithm, makes use of multi-objective user priors via its acquisition function (Section 4.1) and cheap approximations of the objective functions with its initial design (Section 4.2). As `PriMO` yields state-of-the-art for the multi-objective and single-objective setting (Section 5), we discuss how single-objective problems imply a special case of PriMO (Section 4.3). We provide additional details in Appendix C.

## 4.1 INTEGRATING MULTI-OBJECTIVE EXPERT PRIORS INTO BAYESIAN OPTIMIZATION

We first choose one of the priors over multiple objectives uniformly at random during each iteration. We weight the acquisition function of BO (Algorithm 2) with the PDF of the selected prior, raised to an exponent $\gamma = exp\left(-n_{\text{BO}}^2/n_d\right)$. Unlike $\pi$BO, where $\gamma = \frac{10}{n}$, with $n$ referring to the $n^{\text{th}}$ iteration, we reduce the overdependence on the prior by setting $\gamma$ to be inversely proportional to the square of the number of BO samples. To formulate an acquisition function, we convert the vector-valued objective function into a single-objective optimization problem, using a linear scalarization function (Yoon et al., 2009) with randomly sampled weights, which is not only simple but also scalable with the number of objectives $n$:

$$\min_{\lambda \in \Lambda} \sum_{i=1}^{n} w_i \hat{f}_i(\lambda, z_{max}) \qquad w_i \sim \mathcal{U},\ w_i > 0,\ \sum_{i=1}^{n} w_i = 1\,. \tag{3}$$

Furthermore, to aid in recovery from misleading priors, we incorporate a simple exploration parameter $\epsilon$, which controls how often we augment the acquisition function with the prior. Thus, for `PriMO`'s $\epsilon$-BO, with priors over $n$ objectives, the acquisition function becomes

$$\alpha_{\epsilon\pi}(\lambda, \mathcal{D}) \triangleq \begin{cases} \alpha(\lambda, \mathcal{D})\,, & \text{with prob. } \epsilon \\ \alpha(\lambda, \mathcal{D}) \cdot \pi_{f_j}(\lambda)^{exp\left(-n_{\text{BO}}^2/n_d\right)}\,, & \text{with prob. } 1 - \epsilon,\ j \sim \mathcal{U}(1, \dots, n)\,. \end{cases} \tag{4}$$

## 4.2 AN INITIAL DESIGN TO UTILIZE CHEAP APPROXIMATIONS

To leverage cheap approximations of the objective functions, we propose an initial design strategy (Algorithm 1) that exploits the strengths of multi-fidelity algorithms. Specifically, we use a multi-fidelity algorithm in `PriMO` to generate strong initial seed points at the maximum fidelity $z_{max}$ to speed up the optimization in the BO phase afterward.

First, we set a threshold of (equivalent) full function evaluations based on the initial design size. Once this threshold is reached, only maximum fidelity evaluations $\{(\lambda, \hat{f}(\lambda, z_{max})\}$ are included in the dataset $\mathcal{D}$ for use in BO. Next, we choose one of the priors over multiple objectives uniformly at random during each iteration and the sampled initial points then aid the BO along with the decaying prior-augmented acquisition function (Equation 4). We chose to use multi-objective asynchronous successive halving (MOASHA) in our initial design due to its strong performance early on and since as an infinite-horizon optimizer, it is budget invariant, resulting in a single continued optimization run without being restricted to discrete Successive Halving brackets.

**Algorithm 1** Initial design strategy

1: **function** init($n_{\text{init}}, \Lambda, \eta, z_{min}, z_{max}, f, \mathbf{w}$)
2:     $b \leftarrow 0, \mathcal{D} \leftarrow \emptyset$
3:     **while** $b < n_{\text{init}}$ **do**
4:         $\lambda, z \leftarrow$ moasha($\Lambda, \eta, z_{min}, z_{max}$)
5:         $\mathbf{y} \leftarrow f(\lambda, z)$
6:         **if** $z = z_{max}$ **then**
7:             $\mathbf{y} \leftarrow \mathbf{w}^{\top} \mathbf{y}$
8:             $\mathcal{D} \leftarrow \mathcal{D} \cup \{(\lambda, \mathbf{y})\}$
9:         $b \leftarrow b + \frac{z}{z_{max}}$
10:     **return** $\mathcal{D}$

**Algorithm 2** BO step with multi-obj. priors

1: **function** moprior_bo($\Lambda, \mathcal{D}, \Pi_f, n_{\text{BO}}, \epsilon$)
2:     Select prior $\pi_{f_j}$, where $j \sim \mathcal{U}(1, \ldots, n)$
3:     $\gamma \leftarrow \exp(-n_{\text{BO}}^2/n_d)$
4:     $u \sim \mathcal{U}(0, 1)$
5:     **if** $u < \epsilon$ **then**
6:         $\tilde{\alpha}(\lambda) := \alpha(\lambda, \mathcal{D})$
7:     **else**
8:         $\tilde{\alpha}(\lambda) := \alpha(\lambda, \mathcal{D}) \cdot \pi_{f_j}(\lambda)^{\gamma}$
9:     $\lambda \leftarrow \arg\max_{\lambda \in \Lambda} \tilde{\alpha}(\lambda)$
10:     **return** $\lambda$

**Algorithm 3** PriMO

1: **Input:** Objective $f$, search space $\Lambda$ with dimension $n_d$, priors $\Pi_f = \{\pi_{f_i}(\lambda)\}_{i=1}^{n}$, initial design size $n_{\text{init}}$, reduction factor $\eta$, fidelity range $[z_{min}, z_{max}]$, budget $B$ and exploration parameter $\epsilon$.
2: **function** PriMO(Input)
3:     Sample weights $\mathbf{w} \sim \mathcal{U}(0, 1)^n$ and normalize
4:     $\mathcal{D} \leftarrow$ init($n_{\text{init}}, \Lambda, \eta, z_{min}, z_{max}, f, \mathbf{w}$)
5:     $b \leftarrow n_{\text{init}}, n_{\text{BO}} \leftarrow 0$
6:     **while** $b < B$ **do**
7:         $\lambda_{new} \leftarrow$ moprior_bo($\Lambda, \mathcal{D}, \Pi_f, n_{\text{BO}}, \epsilon$)
8:         $n_{\text{BO}} \leftarrow n_{\text{BO}} + 1$
9:         $\mathbf{y} \leftarrow f(\lambda_{new}, z_{max})$
10:         $\mathbf{y} \leftarrow \mathbf{w}^{\top} \mathbf{y}$
11:         $\mathcal{D} \leftarrow \mathcal{D} \cup \{(\lambda_{new}, \mathbf{y})\}$
12:         $b \leftarrow b + 1$
13:     **return** $\mathcal{P}_f(\mathcal{D})$

### 4.3 THE SINGLE-OBJECTIVE SETTING

We also adapt PriMO to the single-objective setting as a special case of the original multi-objective design. The initial design strategy of PriMO replaces MOASHA with the ASHA scheduler. Instead of selecting a prior at random, as in the multi-objective case, we sample from the prior over the single objective. The $\epsilon$-greedy prior-augmented Bayesian Optimization phase remains unchanged.

### 4.4 DISCUSSION

**Runtime Analysis** We use scalarization to transform the multi-objective problem to a single-objective one in each BO step, therefore our runtime behaviour corresponds to classic BO. In our $\epsilon$-BO, we use a Gaussian Processes which has the asymptotic time complexity of $\mathcal{O}(n^3)$. We note that the runtime of a single BO step is negligible in comparison to the model evaluation cost (for an example see Table 2).

Table 2: Comparison of the average HPO sampling times of PriMO and BO+RW, and the evaluation cost of language modeling with a large transformer on the 1B word benchmark.

| Algorithm | Avg. HPO sampling time (s) | Avg. model evaluation time (s) |
|---|---|---|
| PriMO | 10.28 $\pm 1$ | 8306.68 |
| BO+RW | 9.80 $\pm 1$ | 8740.58 |

**Behavior under highly correlated priors** When the priors for two objectives are strongly positively correlated, the overall dependence on their quality increases. If the priors are helpful for both objectives, PriMO's performance will improve, while the converse is true if the priors for both objectives are misleading. However, under mixed prior conditions, their

effects may cancel out or the prior over one objective may suppress the effect of the other depending on the strength of the correlation between them.

## 5 EXPERIMENTS

To empirically demonstrate that `PriMO` fulfills the desiderata outlined in the Introduction, we address the following research questions.

**RQ1**: Does `PriMO` outperform strong multi-objective baselines in terms of anytime and final performance?

**RQ2**: Does `PriMO` maintain state-of-the-art performance in the single-objective setting?

**RQ3**: Can `PriMO` effectively leverage multi-objective expert priors?

**RQ4**: Does `PriMO` recover from misleading priors and maintain its robustness?

**RQ5**: Can `PriMO` effectively leverage cheap approximations with its initial design strategy?

**RQ6**: Are all components of `PriMO` necessary and helpful?

After providing details on our experimental setup and baselines (Section 5.1 and 5.2), we show `PriMO`'s state-of-the-art performance in the multi-objective and single-objective setting in Section 5.3 (answering **RQ1**, **RQ2**, and **RQ3**). We then discuss its robustness across all prior conditions in Section 5.4 (answering **RQ4**), and, finally, provide an ablation study in Section 5.6 (answering **RQ5** and **RQ6**). In Appendix I we provide detailed case-level analysis based on Pareto fronts, and hypervolume plots under bad and overall prior combinations.

### 5.1 EXPERIMENTAL SETUP

**Evaluation protocol**   We base our multi-objective evaluation on the mean dominated hypervolume across 25 seeds and report relative rankings of the algorithms across budgets. Each optimizer-benchmark-seed combination was run for 20 equivalent full function evaluations, corresponding to typical budgets in practical Deep Learning. We give more details on our evaluation protocol in Appendix H, provide additional experiments and analysis in Appendix I, and conduct a statistical significance analysis in Appendix J.

**Benchmarks**   We use 8 benchmarks representing image classification, language translation and learning curves for Deep Neural Networks. We chose 4 LCBench benchmarks from the Yahpo-Gym  Suite (Pfisterer et al., 2022) and 4 from the PD1 (Wang et al., 2024) set of benchmarks. We select the corresponding validation error and training cost metrics as objectives. In Appendix G we provide full details on the 8 benchmarks.

**Priors**   We study the effect of different prior conditions: both objectives have good priors, both objectives have bad priors, the average over mixed good and bad prior combinations, as well as the average over all combinations. For generating priors, we follow the protocols in the literature on single-objective optimization (Appendix E).

### 5.2 BASELINES

We give an overview of all our baselines here and provide additional details in Appendix F.

**Multi-objective baselines from the literature**   We compare `PriMO` against a host of prominent MO baselines representing different classes of optimization algorithms for MO. These include scalarized Bayesian Optimization approaches like BO with random weights (BO+RW) and ParEGO (Knowles, 2006), multi-fidelity optimizers such as HyperBand with Random Weights (HB+RW) (Schmucker et al., 2020) and multi-objective asynchronous successive halving (MOASHA) (Schmucker et al., 2021), and an evolutionary algorithm – NSGA-II (Deb et al., 2002).

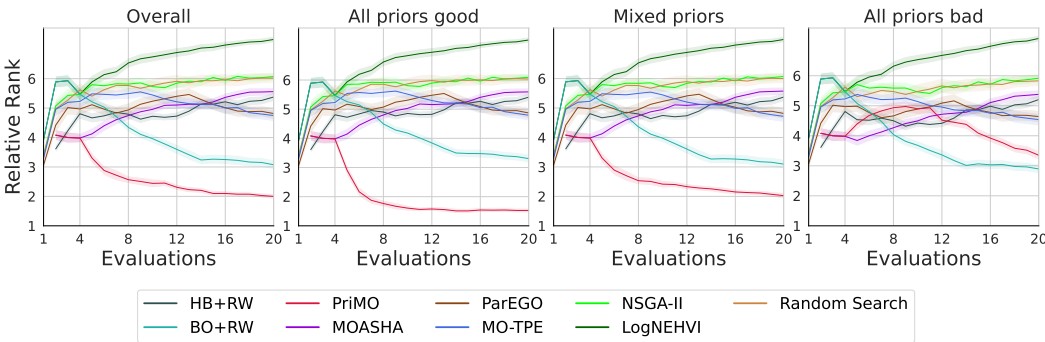

Figure 3: Mean relative ranks $\pm$ 1 standard error of `PriMO` and prominent multi-objective algorithms across benchmarks and seeds under various prior conditions.

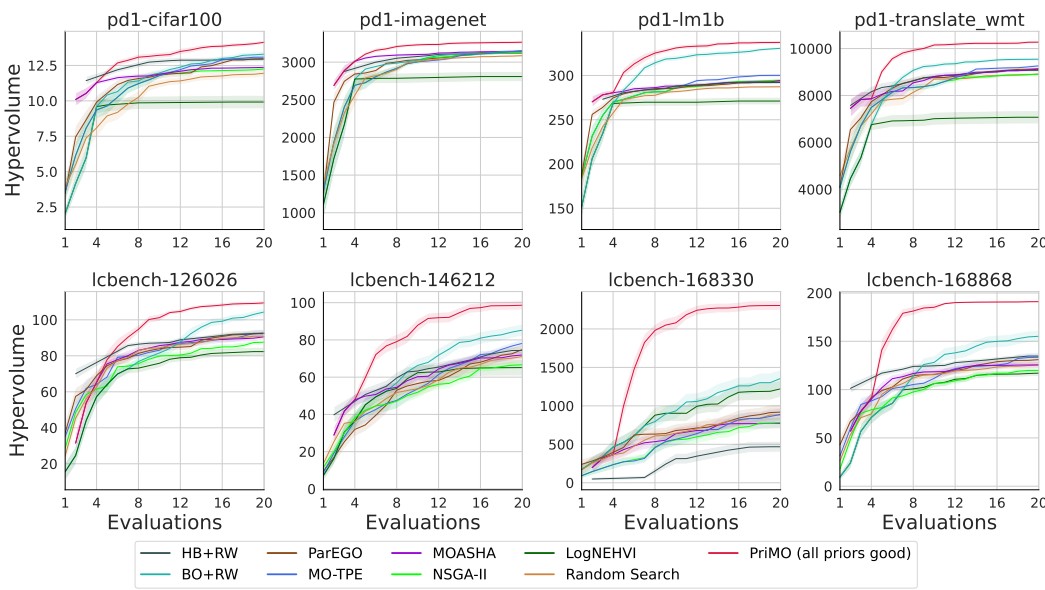

Figure 4: Mean dominated Hypervolume $\pm$ 1 standard error of `PriMO` and prominent multi-objective algorithms across seeds for each benchmark. `PriMO` is under all good priors setting here. See Appendix I for additional Hypervolume plots.

**Additional multi-objective baselines we constructed**   While no multi-objective approaches exist in the literature that leverage expert priors, we augment single-objective approaches to provide strong prior-based baselines. We modify such single-objective approaches (RS + Prior, MOASHA + Prior, $\pi$BO (Hvarfner et al., 2022), Priorband (Mallik et al., 2023)) to randomly chose and sample from one of the multi-objective priors at each iteration. We further augment $\pi$BO with random scalarizations and modify Priorband's ensemble sampling policy using scalarized incumbents for MO to build MO-Priorband.

**Single-objective baselines**   For our experiments in the single-objective setting, we compare `PriMO` against BO and HyperBand (Li et al., 2017), and existing single-objective algorithms that can leverage expert priors, i.e., Priorband-BO and $\pi$BO.

### 5.3   PRiMO ACHIEVES STATE-OF-THE-ART PERFORMANCE

**Multi-objective setting**   Figure 3 demonstrates that, overall, `PriMO` maintains the strongest anytime performance and achieves the best final performance in terms of relative rankings across all benchmarks (**RQ1**). Under good priors the relative ranking gap

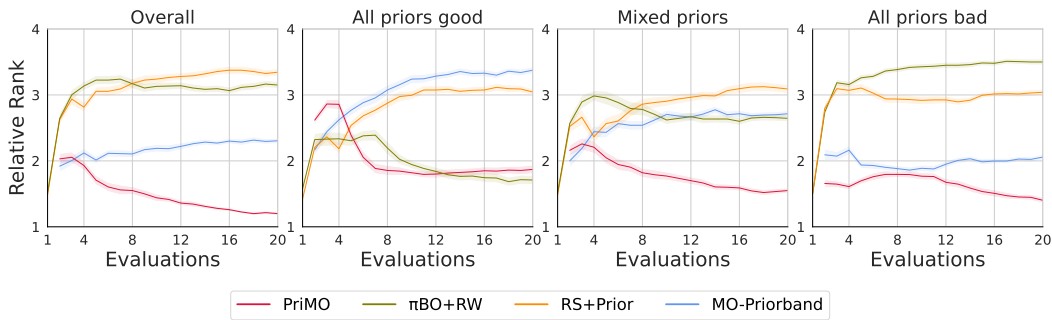

Figure 5: Mean relative ranks $\pm$ 1 standard error of baselines we constructed to use multi-objective priors and `PriMO` across benchmarks and seeds under various prior conditions.

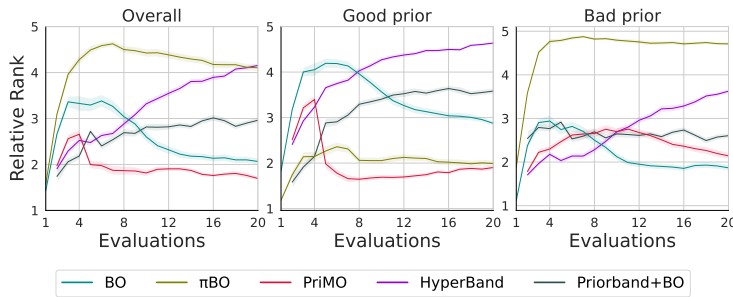

Figure 6: Mean relative ranks $\pm$ 1 standard error in the single-objective setting across benchmarks and seeds under various prior conditions.

between `PriMO` and the second best optimizer, BO+RW, is even more pronounced. `PriMO` shows strong starts across all benchmarks, with respect to the mean dominated Hypervolume under good priors (Figure 4), and is the best performing algorithm across all benchmarks early on. We attribute this to the initial design which provides a strong head-start before the BO phase. Using the configurations sampled by the initial design, the BO phase of `PriMO` maintains its strong anytime performance, and is able to effectively utilize the priors, achieving state-of-the-art final performance across all benchmarks (**RQ3**). Figure 5 clearly shows that, overall, `PriMO` is anytime better compared to prior-based MO adapted baselines and outperforms MO-Priorband by a wide margin. Under good priors, `PriMO` and $\pi$BO+RW are usually 2 of the best optimizers on average. Designed with the practical DL use case in mind, where most practitioners operate on modest budgets, `PriMO` reduces its dependence on priors after approximately 10 BO samples, governed by our chosen $\gamma$ setting (see Section 4).

**Single-objective setting**   Figure 6 shows `PriMO`'s state-of-the-art performance in the single-objective setting. `PriMO` is overall the best choice for single-objective HPO demonstrating the strongest anytime and final performance (**RQ2**). Under good priors, Priorband and $\pi$BO show stronger starts as is expected with their prior-based initial sampling strategies, but are quickly outperformed by `PriMO` within a few full function evaluations.

## 5.4 PRIMO IS ROBUST TO PRIOR CONDITIONS

Under all bad priors in Figure 3 we see that after a poor initial performance, `PriMO` shows remarkable recovery and by the end of the optimization budget, nearly catches up with BO+RW, resulting in a competitive final performance (**RQ4**). Mixed and overall prior conditions show similar trends where `PriMO` is the best performing algorithm early on and ranks significantly better than all other baselines by the final iteration. Compared to MO-adapted prior-based baselines in Figure 5 (all priors bad), we observe that `PriMO` not only has good anytime performance, but also significantly outperforms all baselines by the end of the optimization run across most benchmarks. Thus, $\pi$BO+RW, despite being able to leverage

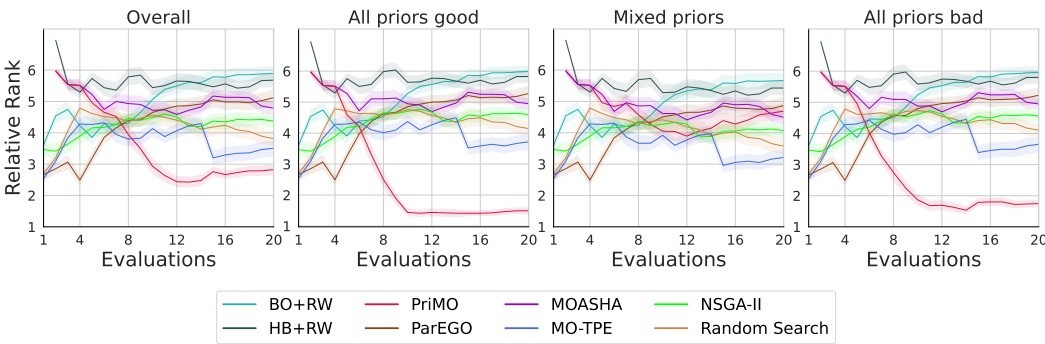

Figure 7: Mean relative ranks ± 1 standard error of `PriMO` and prominent multi-objective algorithms across the 3 HW-GPT-Bench tasks under various prior conditions. For an explanation of `PriMO`'s strong performance under all priors bad, see Section 5.5.

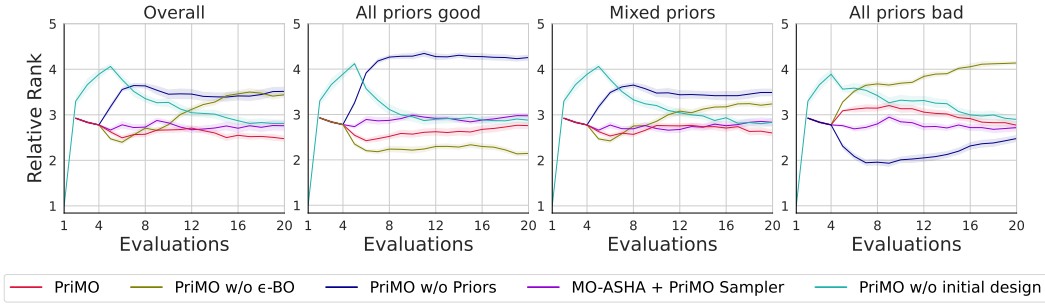

Figure 8: Mean relative ranks ± 1 standard error of various ablations of `PriMO` across benchmarks and seeds under various prior conditions.

good priors well, is quite prone to misleading priors and, overall, significantly (Appendix J) less robust compared to `PriMO`.

## 5.5 CASE STUDY: GPT2

We conduct a case study on the GPT2-based Neural Architecture Search benchmark HW-GPT-Bench (Sukthanker et al., 2024). In Figure 7, we compare the performance of `PriMO` on HW-GPT-Bench against our baselines and find that `PriMO` achieves the strongest performance. We notice that `PriMO` performs surprisingly strong when all priors are misleading for their respective objective. Table 3 shows that a bad prior for perplexity is strongly positively correlated with the FLOPS objective. Therefore, a bad prior for perplexity serves as a good prior for FLOPS.

Table 3: Spearman's rank correlation between objectives and bad priors on HW-GPT-Bench.

|  | Bad prior for FLOPS | Bad prior for perplexity |
|---|---|---|
| FLOPS | -0.06 | 0.56 |
| Perplexity | 0.11 | -0.65 |

## 5.6 ALL COMPONENTS OF PRIMO ARE HELPFUL

We consider different design ablations of `PriMO` in Figure 8 to answer **RQ5** and **RQ6** and, overall, find that all components of `PriMO` are important. We divide `PriMO` into its constituent components, namely – the initial design, MO-Priors and the $\epsilon$-BO (including the random weights), and label each design ablation with respect to the component(s) that were removed from `PriMO`.

We find that the initial design strategy gives a substantial early boost, as all ablations of `PriMO` using the initial design start off much stronger than `PriMO` without the initial design (**RQ5**). This initial advantage tends to persist for most ablations until the BO phase for `PriMO` without the initial design. Overall, while the MOMF initial design provides meaningful early speedups to BO, it does not sustain strong performance in the long run unless paired with the $\epsilon$-BO.

`PriMO` without Priors and `PriMO` without $\epsilon$-BO are two of the worst performing ablations overall, highlighting the importance of a prior-based BO design, coupled with the $\epsilon$-greedy optimization strategy in `PriMO`'s BO. We further notice that MOASHA, when augmented with `PriMO`'s $\epsilon$-BO sampler is surprisingly robust under all prior conditions, although never the most competitive design. These findings, taken together, support our final design choice for `PriMO` (**RQ6**).

## 6 RELATED WORK

Multi-objective optimization traditionally considers large budgets (Deb, 2013; Deb et al., 2002; Knowles, 2006; Golovin & Zhang, 2020); in DL, however, budgets are constrained. Thus, to make HPO for DL feasible, specialized strategies have been proposed. We discuss the strategies most related to ours here and provide a more expansive discussion in Appendix B.

**User priors for single-objective optimization**  The integration of expert priors have been explored in a few works, but only for the single-objective optimization case. Most similar to our approach (albeit for single-objective optimization) is Priorband (Mallik et al., 2023), which, in addition, to exploiting user priors also makes use of cheap approximations, in contrast to us, they use these throughout the optimization and not as an initial design. We explored adapting Priorband to the multi-objective setting, but found it does not perform well (Section 5). $\pi$BO (Hvarfner et al., 2022), like `PriMO`, also augments the acquisition function with the priors, although it does so for a single objective only, can not utilize cheap approximations, and adapting it directly to the MO setting does not perform well under misleading priors (Section 5).

**Exploiting cheap approximations for multi-objective optimization**  While exploiting expert priors is novel for multi-objective HPO, cheap proxies have been explored (Schmucker et al., 2020; Salinas et al., 2021; Schmucker et al., 2021). However, in contrast to our approach, not as an initial design. In our ablation study, we show that integrating cheap approximations as an initial design performs better overall (Section 5.6).

## 7 LIMITATIONS

In line with previous work (Souza et al., 2020; Hvarfner et al., 2022; Mallik et al., 2023), we only consider Gaussian distributions for our priors, where the mean corresponds to a configuration that is expected to perform well, although `PriMO` supports priors with any distribution. While it may be more beneficial in the multi-objective setting to generate priors based on an approximate Pareto front, this remains a non-trivial challenge. However, it is unclear how experts would define priors over a Pareto front directly, and `PriMO` already achieves state-of-the-art performance using simple priors. Additionally, instead of linear scalarization, an approach such as Hypervolume scalarization (Golovin & Zhang, 2020) could be beneficial to `PriMO` as it has provable guarantees to converge to non-convex Pareto fronts.

## 8 CONCLUSION

`PriMO` distinguishes itself as the first algorithm to integrate multi-objective expert priors, leading to state-of-the-art performance. As such, `PriMO` is, to date, the only HPO algorithm that fulfills all the desiderata of modern HPO, making it fit for efficient optimization under constrained budgets for practical Deep Learning.

## Reproducibility statement

To ensure reproducibility, we adhere to and include a reproducibility checklist in Appendix A. In our justifications for the checks, we link to all relevant sections, appendices, and supplementary materials. We considered the checklists in use by NeurIPS and AutoML-Conf, but chose the latter as it is more comprehensive.

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

## B  BACKGROUND AND ADDITIONAL RELATED WORK

### B.1  HYPERPARAMETER OPTIMIZATION FOR DEEP LEARNING

**Multi-fidelity optimization**  The high computational cost of DL model evaluations has motivated research in multi-fidelity optimization. Multi-fidelity (MF) (Kandasamy et al., 2017) optimizers use *cheap proxies* to approximate promising candidates and speed up the search. Bandit-based methods (Jamieson & Talwalkar, 2016; Li et al., 2017) are the most popular in the Automated Machine Learning community for multi-fidelity optimization. These have been further extended by replacing their Random Search (RS) component with evolutionary (Awad et al., 2021) and model-based (Falkner et al., 2018) search, and increasing efficiency for large-scale parallelization (Li et al., 2020).

Instead of optimizing the expensive objective function $f$ as a blackbox, multi-fidelity optimization leverages evaluations of $f$ at lower fidelities. For example, when training a Neural network with a particular hyperparameter configuration for 100 epochs, a lower-fidelity proxy would be the validation score obtained by training the model with the same hyperparameter configuration for 15 epochs. More formally, for a hyperparameter configuration $\lambda \in \Lambda$ at a fidelity level $z \in Z$ where $Z := \{z_{min}, ..., z_{max}\}, |Z| = m$ is the fidelity space, a cheap proxy function of $f$ is defined as $\hat{f}(\lambda, z)$. Therefore, when $z = z_{max}$ (the maximum fidelity), the proxy function $\hat{f}$ converges to the true objective function $f$. Hence, $f = \hat{f}(\lambda, z_{max})$.

In an optimization setup with *continuations*, the function evaluation $\hat{f}(\lambda, z)$ for a configuration $\lambda$ at fidelity $z$ can be continued up to a fidelity $z'$ to yield $\hat{f}(\lambda, z')$, given $z < z'$. For example, let us assume that we would like to train a network with a hyperparameter configuration $\lambda$ for a total of 200 epochs, and have already trained it with $\lambda$ for 50 epochs. Then we can simply continue training with $\lambda$ for 150 more epochs instead of restarting from scratch. For such a continual setup, we define *equivalent function evaluations* as $z/z_{max}$.

**Prior-based optimization for a single objective**  Prior-based single objective optimization can be defined as solving

$$\arg\min_{\lambda \in \Lambda} f(\lambda), \quad \text{guided by } \pi(\lambda), \tag{5}$$

where prior $\pi(\lambda)$ is a probability distribution over the location of the optimum of the objective function $f$.

PrBO (Souza et al., 2020) combines expert prior distributions $P_g(\lambda)$ and $P_b(\lambda)$ with respective models $M_g(\lambda)$ and $M_b(\lambda)$ in a Tree-structured Parzen Estimator (Bergstra et al., 2011) (TPE)-based approach to construct *pseudo-posteriors* $g(\lambda)$ and $l(\lambda)$ respectively. The candidates are then chosen from these pseudo-posteriors by maximizing the EI as described in Bergstra et al. (2011). $\pi$BO (Hvarfner et al., 2022) directly augments the acquisition function $\alpha$ with the unnormalized user-specified prior distribution $\pi(\lambda)$ which decays over time, controlled by a parameter $\beta$: $\alpha_\pi^n(\lambda) = \alpha(\lambda) \cdot \pi(\lambda)^{\frac{\beta}{n}}$, where $n$ refers to the $n^{th}$ iteration. Unlike PrBO, $\pi$BO generalizes to acquisition functions other than EI and offers convergence guarantees. However, as we saw see in Section 5, $\pi$BO's longer dependence on the Priors has major downsides. `PriMO` addresses this issue using a novel MO-priors-based augmentation of the BO component that we introduced in Section 4.

Mallik et al. (2023) introduce Priorband which extends the integration of expert priors to multi-fidelity optimization. Priorband uses a novel *ensemble sampling policy (ESP)* $\mathcal{E}_\pi$, which combines random sampling $\mathcal{U}(\cdot)$, prior-based sampling $\pi(\cdot)$ and incumbent-based sampling $\hat{\lambda}(\cdot)$, with their proportions denoted by $p_\mathcal{U}$, $p_\pi$ and $p_{\hat{\lambda}}$ respectively. Initially, $\hat{\lambda}(\cdot)$ is inactive. Given the constraint $p_\mathcal{U} + p_\pi = 1$, $\mathcal{E}_\pi$ selects from $\hat{\mathcal{U}}(\cdot)$ and $\pi(\cdot)$ according to $p_\mathcal{U}$ and $p_\pi$. When $\hat{\lambda}(\cdot)$ becomes active, $p_\pi$ is split into $p_\pi$ and $p_{\hat{\lambda}}$ according to weighted scores $\mathcal{S}_\pi$ and $\mathcal{S}_{\hat{\lambda}}$, calculated by first computing the likelihood of the top performing configurations under $\pi(\cdot)$ and $\hat{\lambda}(\cdot)$, which capture how much *trust* should be placed on each.

While the aforementioned algorithms efficiently integrate user priors in the HPO problem, they only apply to the single-objective optimization case. To the best of our knowledge, we are the first to incorporate priors over multiple objectives, whilst also employing a novel initial design strategy to leverage cheap proxies of the objective function.

## B.2 MULTI-OBJECTIVE OPTIMIZATION

For many real-world problems we are often interested in optimizing not one, but multiple, potentially competing objectives. MO (Srinivas & Deb, 1994; Deb et al., 2002; Knowles, 2006; Zhang & Li, 2007) deals with optimizing a *vector-valued objective function* $f(\lambda)$ composed of $n$ distinct objective functions, where $f : \lambda \to \mathbb{R}^n$, $\lambda \in \mathbb{R}$. Without loss of generality, we assume minimization of all objectives. More formally, the MO problem can be defined as:

$$\arg \min_{\lambda \in \Lambda} f(\lambda) = \arg \min_{\lambda \in \Lambda} (f_1(\lambda), f_2(\lambda), ..., f_n(\lambda)) \quad . \tag{6}$$

**Pareto optimality** Typically, there does not exist a single best solution for MO problems that minimizes all the objectives simultaneously. Rather, there exists a set of solutions, consisting of points in the domain $\Lambda$.

Given two candidates $\lambda_1, \lambda_2 \in \Lambda$, we say that $\lambda_2$ dominates $\lambda_1$ *if and only if* $f(\lambda_2) < f(\lambda_1)$. Formally, we write $\lambda_2 \prec \lambda_1$. For $f(\lambda_2) \leq f(\lambda_1)$, we write $\lambda_2 \preceq \lambda_1$ and say that $\lambda_1$ weakly dominates $\lambda_2$.

For a vector-valued function $f$, we say that $\lambda_2$ Pareto dominates $\lambda_1$, i.e. $\lambda_2 \prec \lambda_1$ under two conditions:

- $\forall i \in \{1, ..., n\} : f_i(\lambda_2) \leq f_i(\lambda_1)$, and,
- $\exists k \in \{1, ..., n\} : f_k(\lambda_2) < f_k(\lambda_1)$ .

A candidate $\lambda$ that is not dominated by any other candidate $\lambda'$ is called *Pareto Optimal*, and the set of Pareto Optimal candidates is known as the *Pareto Set* $\mathcal{P}$, defined as:

$$\mathcal{P} := \{\lambda \in \Lambda \mid \nexists \lambda' \in \Lambda \text{ with } f(\lambda') < f(\lambda)\} \quad . \tag{7}$$

The set of solutions, i.e., set the corresponding values of an MO function for each of the Pareto Optimal candidates is called the *Pareto Front*. Formally, a Pareto front is defined as:

$$\mathcal{F} = \{f(\lambda) \in \mathbb{R}^n \mid \lambda \in \Lambda, \nexists \lambda' \in \Lambda \text{ with } f(\lambda') < f(\lambda)\} \quad . \tag{8}$$

**Hypervolume indicator**   The true Pareto front of a real-world MO problem is generally unknown. Thus, the goal of MO Optimization algorithms is to return a set of non-dominated candidates from which we can obtain an *approximated Pareto front*. To assess the quality of this approximation, the *S-Metric* or *Hypervolume (HV) Indicator* (Zitzler & Thiele, 1998) is the most frequently used measure as it does not require prior knowledge of the true Pareto front.

Given a reference point $r$ and an approximate Pareto set $\mathcal{A}$, the Hypervolume Indicator $\mathcal{H}$ is defined as:

$$\mathcal{H}_r(\mathcal{A}) = \mu \left( \{ x \in \mathbb{R}^n | \exists a \in A : a \leq x \cap x \leq r \} \right) \quad , \tag{9}$$

where $\mu$ is the Lebesgue measure. Throughout this work, for our experiments, we will be using the *Hypervolume Improvement* (*HVI*) metric as a cumulative performance indicator for MO algorithms with respect to function evaluations. Given a new set of candidates $\gamma$, an existing Pareto set $\mathcal{P}$ and a reference point $r$, the *HVI* is formally defined as:

$$HVI(P, r, \gamma) = \mathcal{H}_r(P \cup \gamma) - \mathcal{H}_r(P) \quad . \tag{10}$$

### B.3   Multi-objective optimization for Deep Learning

For DL, it is often necessary to optimize not only the validation error (or validation accuracy) but also a cost metric, such as the inference time of a Neural Network or Floating Point Operations per Second. It is easy to imagine that a cost metric would be cheap to evaluate since it is a simple observation, unlike an objective such as accuracy (which would require the network to be trained first) (Izquierdo et al., 2021). Additionally, we might also be interested in a third objective like fairness or interpretability of the DL model. However, from a DL perspective, optimizing predictive performance typically (but not always) comes at the cost of degrading other objectives. In the context of Machine Learning, multi-objective algorithms for HPO (Jin & Kacprzyk, 2006) have been adapted mainly from the general MO literature.

**Scalarization-based Bayesian Optimization**   Scalarization-based multi-objective Bayesian Optimization (MO-BO) approaches (Knowles, 2006; Golovin & Zhang, 2020; Paria et al., 2019; Yoon et al., 2009) use a function: $s : \mathbb{R}^n \times \alpha \mapsto \mathbb{R}$ that maps the vector-valued MO function into a scalar value, thus effectively converting the MO problem into a single-objective optimization problem. These approaches vary in the choice of the scalarization function (Knowles, 2006) or the distribution from which the weights are sampled (Yoon et al., 2009; Paria et al., 2019). Knowles (2006) introduced *ParEGO*, which uses a Tchebycheff norm over the objective values as opposed to a linear weighted sum approach. These methods are highly scalable and easy to implement, which is why we employ random scalarizations during the BO phase of `PriMO`.

**Multi-objective Bayesian Optimization using acquisition function modifications**   Other MO-BO approaches directly modify the acquisition function in BO to account for multiple objectives. Emmerich (2005) proposed the Expected Hypervolume Improvement (EHVI) acquisition function wherein a surrogate model is fitted for each objective separately, and then the Expected Improvement (EI) (Jones et al., 1998) of the HV contribution is calculated. Several improvements to calculate EHVI have been proposed, such as in Yang et al. (2019) and Daulton et al. (2020). EHVI is also used in Ozaki et al. (2020) to extend the TPE (Bergstra et al., 2011) to MO TPE. Ponweiser et al. (2008) introduced the S-Metric Selection-based EGO (SMS-EGO) which, instead of using EHVI, selects new candidates by directly maximizing the HV contribution based on the predictions of the surrogate model, using the Lower Confidence Bound (Jones, 2001) acquisition function. Izquierdo et al. (2021) modified EHVI by fitting surrogate models only on the expensive objectives, such as validation accuracy. MO Information-theoretic acquisition functions, such as maximum entropy search (Belakaria et al., 2019) (MESMO) and predictive entropy search (Hernández-Lobato et al., 2016) (PESMO), aim to reduce the entropy of the location of the Pareto front.

**Evolutionary algorithms**   Evolutionary MO Algorithms mutate configurations from a diverse initial population to identify promising candidates closer to the Pareto front. Deb et al. (2002) proposed the popular Non-dominated Sorting Genetic Algorithm (NSGA-II) which

uses non-dominated sorting (Srinivas & Deb, 1994) to rank candidates from multiple non-dominated fronts and conducts survival selection (tie-breaking) using crowding-distance sort (Deb et al., 2002). S-Metric Selection Evolutionary Multi-objective Optimization Algorithm (SMS-EMOA) (Beume et al., 2007) also employs non-dominated sorting from (Srinivas & Deb, 1994) and (Deb et al., 2002) for the initial ranking of candidates, but then uses each candidate's contribution to the dominated HV for survival selection. Evolutionary methods, however, are quite compute-inefficient, requiring a high budget to significantly improve the dominated HV. Compute efficiency is one of the desiderata we identify in Table 1 and therefore is a key aspect of `PriMO`.

**Multi-objective multi-fidelity optimization** Izquierdo et al. (2021) extended SMS-EMOA to the MF domain by augmenting it with SH rungs. Furthermore, they introduced MO-BOHB, which replaced the TPE component with MO-TPE. Schmucker et al. (2020) adapted HyperBand (HB) to MO using a randomly scalarized objective value (HB+RW) to select and promote promising configurations. Salinas et al. (2021) and Schmucker et al. (2021) further build on Schmucker et al. (2020) by modifying the promotion strategy of HB and ASHA respectively, using non-dominated sorting for the initial ranking of candidates, and a greedy `epsilon-net` ($\epsilon$-net) strategy for exploration.

MF-OSEMO (Belakaria et al., 2020b) and iMOCA (Belakaria et al., 2020a) extend the information-theoretic method MESMO to discrete and continuous fidelities, respectively. Irshad et al. (2024) propose a novel modification to the EHVI acquisition function which optimizes a multi-objective function and the fidelity of the data source jointly. They achieve this by defining a *trust-based cost objective* which is directly proportional to the fidelity level. However, these MOMF-BO algorithms are quite computationally expensive, requiring vast amounts of resources and longer optimization runtimes. Although they integrate cheap approximations of the objective function, their high overall computational costs make them unsuitable for DL.

Apart from a few notable exceptions, MO algorithms have been largely been used for general optimization problems. Their usage in practical DL applications have been relatively limited compared to single-objective optimization, and only a handful of studies exist where MO optimizers are benchmarked on real-world DL tasks. We aim to bridge this gap between general multi-objective optimization and multi-objective hyperparameter optimization by demonstrating `PriMO`'s effectiveness in both synthetic MO problems, as well as DL benchmarks.

## C  ALGORITHM DETAILS

All the parts of `PriMO` were implemented based on the NePS (Stoll et al., 2025) package. For `PriMO`'s initial design strategy we used MOASHA already implemented in NePS. MOASHA in NePS uses the $\epsilon$-net MO promotion strategy from the Syne Tune repository, which is the original implementation by its authors (Schmucker et al., 2021). It is also important to note here that MOASHA is the most viable choice for the initial design compared to bandit-based optimizers with synchronous promotions like HB or SH. This is because the latter would require much longer budgets to promote configurations to the highest fidelity rung, which is impractical for the initial design size of BO. We set $\eta = 3$ and the initial design size to 5. Furthermore, we set $\epsilon = 0.25$ in our experiments, selecting the prior-weighted acquisition with a probability of 0.75.

In Figure 9, we plot the decay of $\gamma$, the exponent of the prior in the weighted acquisition function, in case of `PriMO` and the original $\pi$BO implementation, with increasing number of Bayesian Optimization samples. We show that the $\gamma$, and hence the strength of the prior, decays much faster in `PriMO` in comparison to $\pi$BO, which helps to recover from misleading prior information much quicker.

We use the BoTorch implementation of `SingleTaskGP` as s surrogate model for purely continuous or purely discrete spaces, and `MixedSingleTaskGP` for mixed search spaces. For the base acquisition function in the prior-augmented BO, we used `qLogNoisyEI` from BoTorch as it has been proven to significantly outperform ordinary EI implementations (Ament et al.,

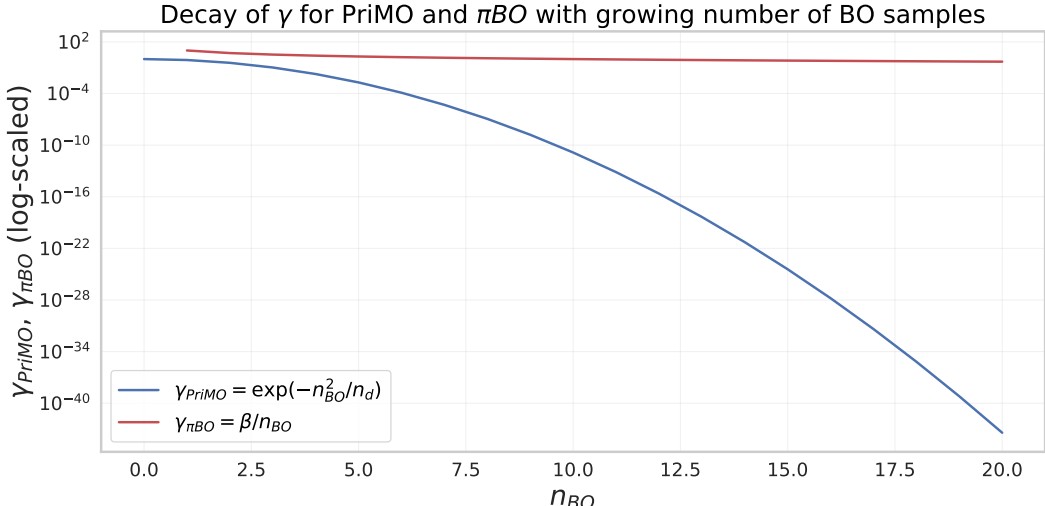

Figure 9: Decay of the prior exponent $\gamma$ with the increase in the number of Bayesian Optimization samples for `PriMO` and the original $\pi$BO implementation. Here, $n_d$ refers to the search spaces dimensions, which we fix to 4 - the size of the search space of the `PD1` benchmarks. We note that the $\pi$BO authors (Hvarfner et al., 2022) set $\beta = 10$ in the paper.

2023). The NePS package already contains code for the `WeightedAcquisition` function for $\pi$BO, which we borrow for `PriMO`.

## D  Sources of priors

In practice, Deep Learning experts often have substantial intuition about ranges of hyperparameter settings that work best for certain models and downstream tasks (Souza et al., 2021; Hvarfner et al., 2022; Mallik et al., 2023). For example, an ideal learning rate for the Adam optimizer (Kingma & Ba, 2015) is often around $1 \times 10^{-3}$ (Hvarfner et al., 2022). Prior knowledge often stems from the literature, experience in tuning similar models, repeated experimentation or transfer learning across tasks. Consequently, DL researchers often start their experiments with hyperparameter settings that are reported in the original paper for that architecture. In recent literature, large language models have also been used to generated priors for Hyperparameter Optimization (Zhang et al., 2023). Priors for resource-related objectives, such as training time, FLOPS, memory consumption, or energy usage, are generally easier to define, as these objectives are directly observable and can hence be easily measured.

## E  Construction of priors

For the construction of priors, we closely follow the procedure described by Mallik et al. (2023). Our priors are hyperparameter settings, perturbed by a Gaussian noise with a $\sigma$ depending on the prior quality. In all our experiments we use two kinds of priors for every objective - *good* and *bad* priors. The good priors represent areas of the hyperparameter space where we expect the corresponding objective to have a value close to its *optimum*. The bad priors represent *inaccurate* configurations which yield poor values for the objective function. The hyperparameter configurations for these priors are generated using the methods listed below:

- Class "**good**" priors: To generate *good* priors, we begin by uniformly sampling 100,000 hyperparameter configurations at random using a fixed global seed for all prior generation runs. We then evaluate these configurations on the corresponding benchmark at the highest available fidelity, $z_{max}$. Afterwards, we rank the config-

urations based on the objective values derived from their evaluations. Since we always aim to minimize each objective, for objectives intended to be maximized, we take their negative values to find the minimum. The configuration that yields the best objective value is perturbed by a Gaussian noise with $\sigma = 0.01$. This slight perturbation reflects a realistic scenario where prior knowledge is good or near-optimal, but never precisely so.

- Class "**bad**" priors: Similar to the good prior case, for the *bad* priors, we sort the configurations based on the corresponding objective value. From this, we select the configuration with the worst seen value and do not perturb it any further. This forms our *bad* prior configuration.

After locating the hyperparameter configurations that constitute these priors, we create a Gaussian distribution over each, $\mathcal{N}(\lambda, \sigma^2)$, where $\sigma = 0.25$ for all priors.

## F  BASELINES

The implementation and hyperparameter setting of all baselines used in this paper are individually detailed below

### F.1  SINGLE-OBJECTIVE BASELINES

**Bayesian Optimization (BO)**   Bayesian Optimization is a popular HPO algorithm that builds a probabilistic model to estimate the optimum of a blackbox objective function. We select the Gaussian Processes-based BO implementation from the NePS (Stoll et al., 2025) package, which uses the `q-Log-Noisy Expected Improvement` acquisition function from BoTorch. This has been shown to perform significantly better than ordinary Expected Improvement implementations. The initial design size of the BO is set to be the same as the dimensionality of the corresponding benchmark's search space (Ament et al., 2023).

**HyperBand (HB)**   HyperBand (Li et al., 2017) is a common multi-armed bandit-based HPO algorithm that iterates over multiple Successive Halving brackets, and is a common baseline MF benchmarking studies. The NePS package provides an implementation of HB which allows for continuations, and we set $\eta = 3$ for all our experiments.

**$\pi$BO**   $\pi$BO is a single-objective, blackbox optimization algorithm which augments the acquisition function with user-specified priors. We use the $\pi$BO implementation from the NePS package. The original $\pi$BO paper (Hvarfner et al., 2022) uses $\gamma = \frac{\beta}{n}$ to denote the power to which the prior PDF term is raised when multiplied by the values of the acquisition function, where $n$ refers to the $n$-th iteration and the value of $\beta$ is set to 10. In the NePS package, however, $\gamma$ is completely different and is set to $e^{-n_{BO}/n_d}$, where $n_{BO}$ refers to the number of BO samples and $n_d$ indicates the dimensions of the search space.

### F.2  MULTI-OBJECTIVE BASELINES FROM THE LITERATURE

**Bayesian Optimization with Random Weights (BO+RW)**   BO with random weights is a popular MO baseline which converts the MO function into a SO optimization problem. Keeping all the settings as described in BO above, we extend the BO implementation in the NePS package by scalarizing the multivariate objective function $f$ with randomly chosen weights for every seed, at the beginning of the optimization process.

**ParEGO**   Just like BO+RW, ParEGO (Knowles, 2006) is another BO baseline with the Chebyshev norm as the scalarization function. We use the ParEGO implementation from the SMAC3 package and leave the initial design design size of the BO as the package default (search space dimensions).

**Bayesian Optimization with Expected Hypervolume Improvement acquisition function (LogNEHVI)**   The Expected Hypervolume Improvement acquisition function

was introduced by (Emmerich, 2005) which fits a surrogate model separately for each objective of the multi-objective problem. We employ the BoTorch implementation of Log Noisy Expected Hypervolume Improvement (Ament et al., 2023) from the NePS package. The initial design size for LogNEHVI is set to the size of the search space.

**Multi-objective Tree-structured Parzen Estimator (MO-TPE)**  MO-TPE (Ozaki et al., 2020; 2022) is an extension of the popular Tree-structure Parzen Estimator algorithm using EHVI as an acquisition function. We use the Optuna implementation of MO-TPE with recent computational and performance improvements (Abe et al., 2025). Like the other BO algorithms, we set the initial random sampling of Optuna's MO-TPE to the number of dimensions of the search space.

**NSGA-II**  NSGA-II (Deb et al., 2002) is an EA algorithm which uses non-dominated sorting to identify promising configurations and crowding-distance sort as a tie-breaker. It is a popular baseline but EAs are quite sample-inefficient and hence not super practical for DL as a standalone optimization algorithm. Thus, we use NSGA-II as a representative EA baseline and borrow its implementation from the Nevergrad package. The parameters of the algorithm are set to the defaults values defined in Nevergrad.

**HyperBand with Random Weights (HB+RW)**  Following Schmucker et al. (2020), we modify HB from NePS with random weights the same way as BO+RW above. For all our experiments, we set the $\eta = 3$.

**Multi-objective Aynchronous Successive Halving (MOASHA)**  MOASHA is an infinite horizon MO optimizer and currently one of the state-of-the-art baselines for multi-objective optimization, using bandit-based ASHA as the base. Like ASHA, MOASHA can also run very efficiently on HPO setups with many parallel workers, reducing idle-time. However, even for single worker setups, MOASHA is able to leverage its asynchronous promotion strategy to achieve competitive performance (Schmucker et al., 2021), and that is what we employ for the experiments in this paper. We used MOASHA from the NePS package, and just like HB+RW above, we set $\eta = 3$.

### F.3 Baselines we constructed to leverage multi-objective expert priors

**$\pi$BO with random weights ($\pi$BO+RW)**  $\pi$BO+RW is a MO direct extension of $\pi$BO from NePS with random weights, just like BO+RW above. For use with MO priors, we modify $\pi$BO to randomly chose and sample from one of the MO priors at each iteration. Like BO+RW, we set the initial design size to $n_d$, and sample from a randomly chosen prior for each of the initial points. The remaining details of the base $\pi$BO algorithm is the same as detailed above.

**Priorband+BO**  Priorband integrates cheap proxies unlike $\pi$BO to achieve good anytime performance. It employs an ESP strategy for sampling proportionately from the *priors*, the *incumbent* and at *random*. Priorband+BO is a model-based extension of Priorband using Gaussian Processes with the EI acquisition function. NePS provides an implementation of Priorband+BO which we use for our single-objective experiments. Just like in BO above, Priorband+BO uses the `q-Log-Noisy Expected Improvement` acquisition function from BoTorch. Further details about this model-based extension is available in the Priorband paper (Mallik et al., 2023).

**MO-Priorband**  We extend Priorband to the MO domain by first replacing the MF component with an MOMF component. Then, to calculate the $top\_k$ configurations, we scalarize the MO vectors using weights, randomly chosen during each iteration. Additionally, to integrate MO priors, MO-Priorband chooses one of the available priors at random at each iteration. We note that a scalarization-based incumbent modification works better for MO-Priorband than a Pareto front incumbent such as $\epsilon$-net. Additionally, we set MO-Priorband's $\eta = 3$, just as in Priorband.

**Random Search + Prior (RS + Prior)** We first randomly select one of the multi-objective priors to sample from. We then equip Random Search to draw random samples from the selected prior distribution (instead of the entire search space) to construct the RS + Prior baseline.

**MOASHA + Prior** We equip MOASHA to sample from one of the multi-objective prior distributions in the same way as Random Search above. For MOASHA + 50% prior sampling, we choose uniformly at random, between prior-based sampling and random sampling from the entire search space.

## G  BENCHMARKS

Our main experiments in Section 5 include the surrogate benchmarks LCBench-126026, LCBench-146212, LCBench-168330, LCBench-168868 from Yahpo-Gym  and cifar-100, imagenet, translate-wmt-xformer, lm1b-transformer from the PD1 suite.

### G.1  PD1 (HYPERBO)

PD1 from HyperBO (Wang et al., 2024) is a collection of XGBoost surrogates trained on the learning curves of near state-of-the-art DL models on a diverse array of practical downstream DL tasks including image classification, language modeling and language translation. Overall, PD1 contains 24 benchmarking tasks, with each consisting of a task dataset, a DL model, and a broad search space for Nesterov Momentum (Nesterov, 1983).

From these 24, we select 4 benchmarks from `mf-prior-bench` (Bergman et al., 2025) providing a well-rounded representation of DL models and the aforementioned tasks. For each of these benchmarks, we select the `valid_error_rate` as the *validation error* objective and `train_cost` as the *training cost* objective. All of these benchmarks have a single fidelity `epoch`. We list the static reference points for calculating the HVI for the PD1 benchmarks in Table 4. The individual benchmarks are further detailed below:

1. **cifar100-wide_resnet-2048** benchmark contains the optimization trace of a `WideResnet` (Zagoruyko & Komodakis, 2016) model on the `CIFAR-100` (Krizhevsky, 2009) dataset with a batch size of 2048. The hyperparameter space of this benchmark is given in Table 5.

2. **imagenet-resnet-512** surrogate is trained on the learning curve of a `ResNet50` (He et al., 2016) on the `ImageNet` (Russakovsky et al., 2015) dataset with a batch size of 512. See Table 6 for the detailed search space of this benchmark.

3. **lm1b-transformer-2048** is a surrogate trained on the HPO runs of a transformer model (Roy et al., 2021) on the *One Billion Word* statistical language modeling benchmark (Chelba et al., 2014). Table 7 lists the search space of the benchmark.

4. **translatewmt-xformer-64** surrogate is trained on the HPO runs of an `xformer` (Lefaudeux et al., 2022) transformer model on the `WMT15 German-English` text translation dataset (Bojar et al., 2015). For the detailed search space, see Table 8.

Table 4: Reference values for `valid_error_rate` and `train_cost` objectives across PD1 benchmarks for HVI calculation.

| Benchmark Name | valid_error_rate (max) | train_cost (max) |
|---|---|---|
| cifar100-wide_resnet-2048 | 1.0 | 30 |
| imagenet-resnet-512 | 1.0 | 5000 |
| lm1b-transformer-2048 | 1.0 | 1000 |
| translatewmt-xformer-64 | 1.0 | 20000 |

Table 5: Hyperparameter search space table of the `cifar-100-wide_resnet-2048` benchmark, including the hyperparameter ranges and fidelity bounds of `epoch`, as given in `mf-prior-bench` .

| Hyperparameter | Type | Log-scaled | Range | Space Type | Notes |
|---|---|:---:|:---:|:---:|:---:|
| `lr_decay_factor` | float | | $[0.010093, 0.989012]$ | continuous | |
| `lr_initial` | float | ✓ | $[0.000010, 9.779176]$ | continuous | |
| `lr_power` | float | | $[0.100708, 1.999376]$ | continuous | |
| `opt_momentum` | float | ✓ | $[0.000059, 0.998993]$ | continuous | |
| `epoch` | integer | | $[1, 52]$ | discrete | fidelity |

Table 6: Approximate hyperparameter search space table of the `imagenet-resnet-512` benchmark, including hyperparameter ranges and fidelity bounds of `epoch`. Exact ranges are provided by `mf-prior-bench` .

| Hyperparameter | Type | Log-scaled | Range | Space Type | Notes |
|---|---|:---:|:---:|:---:|:---:|
| `lr_decay_factor` | float | | $[0.010294, 0.989753]$ | continuous | |
| `lr_initial` | float | ✓ | $[1e-5, 9.774312]$ | continuous | |
| `lr_power` | float | | $[0.100225, 1.999326]$ | continuous | |
| `opt_momentum` | float | ✓ | $[5.9e-5, 0.998993]$ | continuous | |
| `epoch` | integer | | $[1, 99]$ | discrete | fidelity |

Table 7: Hyperparameter search space of the `lm1b-transformer-2048` benchmark, with the fidelity `epoch` as given in `mf-prior-bench` .

| Hyperparameter | Type | Log-scaled | Range | Space Type | Notes |
|---|---|:---:|:---:|:---:|:---:|
| `lr_decay_factor` | float | | $[0.010543, 0.9885653]$ | continuous | |
| `lr_initial` | float | ✓ | $[1e-5, 9.986256]$ | continuous | |
| `lr_power` | float | | $[0.100811, 1.999659]$ | continuous | |
| `opt_momentum` | float | ✓ | $[5.9e-5, 0.9989986]$ | continuous | |
| `epoch` | integer | | $[1, 74]$ | discrete | fidelity |

Table 8: Search space and fidelity `epoch` of the `translatewmt-xformer-64` benchmark, as given in `mf-prior-bench` .

| Hyperparameter | Type | Log-scaled | Range | Space Type | Notes |
|---|---|:---:|:---:|:---:|:---:|
| `lr_decay_factor` | float | | $[0.0100221257, 0.988565263]$ | continuous | |
| `lr_initial` | float | ✓ | $[1.00276e-5, 9.8422475735]$ | continuous | |
| `lr_power` | float | | $[0.1004250993, 1.9985927056]$ | continuous | |
| `opt_momentum` | float | ✓ | $[5.86114e-5, 0.9989999746]$ | continuous | |
| `epoch` | integer | | $[1, 19]$ | discrete | fidelity |

### G.2 LCBench surrogate benchmarks (Yahpo-Gym)

Yahpo-Gym (Pfisterer et al., 2022) is a large collection of multi-objective multi-fidelity surrogate benchmarks trained on a wide array of tasks with fidelities including epochs as well as dataset fractions. Yahpo-Gym also contains surrogates for the LCBench (Zimmer et al., 2021) set of benchmarks that consists of surrogates trained on the learning curves of DL models, on several OpenML (Vanschoren et al., 2014) datasets. Out of these, we choose 4 task OpenML IDs for the experiments in this paper – 126026, 146212, 168330 and 168868. The fidelity for these tasks is `epoch` and we select the `val_cross_entropy` and `time` as the *validation error* and the *training cost* objectives respectively, for our experiments. Table 9

lists the maximum bounds used as the reference points for calculating the Hypervolume Improvement, for each of the selected LCBench task IDs. All LCBench benchmarks share a common search space, detailed in Table 10.

Table 9: Reference values for `val_cross_entropy` and `time` objectives across selected LCBench tasks, for HVI calculation.

| Task ID | val_cross_entropy (max) | time (max, seconds) |
|---------|-------------------------|---------------------|
| 126026  | 1.0                     | 150                 |
| 146212  | 1.0                     | 150                 |
| 168330  | 1.0                     | 5000                |
| 168868  | 1.0                     | 200                 |

Table 10: Hyperparameter search space table of the `yahpo-lcbench` benchmarks. This includes the hyperparameter ranges and types as typically defined in the YAHPO-Gym benchmark suite.

| Hyperparameter | Type | Log-scaled | Range | Space Type | Notes |
|----------------|------|------------|-------|------------|-------|
| batch_size     | integer | ✓ | $[16, 512]$ | discrete | |
| learning_rate  | float | ✓ | $[1e{-}4, 0.1]$ | continuous | |
| momentum       | float |   | $[0.1, 0.99]$ | continuous | |
| weight_decay   | float |   | $[1e{-}5, 0.1]$ | continuous | |
| num_layers     | integer |  | $[1, 5]$ | discrete | |
| max_units      | integer | ✓ | $[64, 1024]$ | discrete | |
| max_dropout    | float |   | $[0.0, 1.0]$ | continuous | |
| epoch          | integer |  | $[1, 52]$ | discrete | fidelity |

### G.3 Hardware-aware architecture benchmark for language models - HW-GPT-Bench

HW-GPT-Bench (Sukthanker et al., 2024) is a hardware-aware language modeling benchmark for multi-objective NAS. It is based on the GPT-2 architecture with up to ∼1.55 billion parameters. HW-GPT-Bench consists of trained surrogates to predict perplexity and other hardware metrics across multiple devices and architecture scales.

To build the benchmark, the authors train a supernet covering many possible sub-architectures and inherit pretrained weights from the largest GPT-2 model. They use Autogluon as a surrogate model and train it on 100000 randomly sampled architectures from each of the various search space scales to predict the performance and hardware metrics. For the experiments in this paper, we select the **s**, **m** and **l** architecture scales as distinct tasks, and the `perplexity` and `FLOPS` metrics as objectives. For the detailed search spaces, see Table 11, Table 12 and Table 13. The reference points for Hypervolume calculation are listed in Table 14.

## H Details on evaluation protocol

**Computing hypervolume** We compute the HV with respect to a static reference point set for each benchmark (Appendix G).

**Equivalent function evaluations** For blackbox optimizers like BO+RW, NSGA-II and ParEGO, every optimization iteration is equal to a function evaluation since they evaluate $f$ at the maximum fidelity $z_{max}$. For optimizers such as MOASHA, HB+RW and `PriMO` that use cheap proxies of the objective, we calculate equivalent function evaluations as $z/z_{max}$ where $z$ is the fidelity at which $f$ is evaluated at a given iteration. We note here that for all MF optimizers, we leveraged continuations and plot the HV only when an equivalent full

Table 11: Hyperparameter search space for HW-GPT-Bench-s benchmark including fidelity `n_layer_choices`. We note here that HW-GPT-Bench itself does not define a fidelity. We select `n_layer_choices` as a practical fidelity for a NAS benchmark, which scales the `mlp_ratio_choices` and `n_head_choices` parameters accordingly.

| Hyperparameter | Type | Range | Space Type | Notes |
|---|---|---|---|---|
| embed_dims | categorical | $[192, 384, 768]$ | discrete | |
| mlp_ratio_choices | categorical | $[2, 3, 4]$ | discrete | |
| n_head_choices | categorical | $[4, 8, 12]$ | discrete | |
| bias_choices | categorical | $[True, False]$ | discrete | |
| n_layer_choices | integer | $[10, 11, 12]$ | discrete | fidelity |

Table 12: Hyperparameter search space for HW-GPT-Bench-m benchmark including fidelity `n_layer_choices`.

| Hyperparameter | Type | Range | Space Type | Notes |
|---|---|---|---|---|
| embed_dims | categorical | $[256, 512, 1024]$ | discrete | |
| mlp_ratio_choices | categorical | $[2, 3, 4]$ | discrete | |
| n_head_choices | categorical | $[8, 12, 16]$ | discrete | |
| bias_choices | categorical | $[True, False]$ | discrete | |
| n_layer_choices | integer | $[22, 23, 24]$ | discrete | fidelity |

Table 13: Hyperparameter search space for HW-GPT-Bench-l benchmark including fidelity `n_layer_choices`

| Hyperparameter | Type | Range | Space Type | Notes |
|---|---|---|---|---|
| embed_dims | categorical | $[320, 640, 1280]$ | discrete | |
| mlp_ratio_choices | categorical | $[2, 3, 4]$ | discrete | |
| n_head_choices | categorical | $[8, 16, 20]$ | discrete | |
| bias_choices | categorical | $[True, False]$ | discrete | |
| n_layer_choices | integer | $[34, 35, 36]$ | discrete | fidelity |

Table 14: Reference values for `val_cross_entropy` and `time` objectives for HVI calculation across the 3 selected HW-GPT-Bench tasks.

| Task ID | perplexity (max) | FLOPS (max) |
|---|---|---|
| s | 100 | $9 \times 10^{12}$ |
| m | 100 | $9 \times 10^{12}$ |
| l | 100 | $9 \times 10^{12}$ |

function evaluation has been performed, *i.e.*, when the benchmark is evaluated at its highest fidelity $z_{max}$.

**Single-objective evaluations**  We report the relative rankings of all optimizers over all benchmarks based on the normalized regret per benchmark. Similar to the multi-objective case, we run each optimizer-benchmark pair for 20 equivalent full function evaluations, across 25 random seeds.

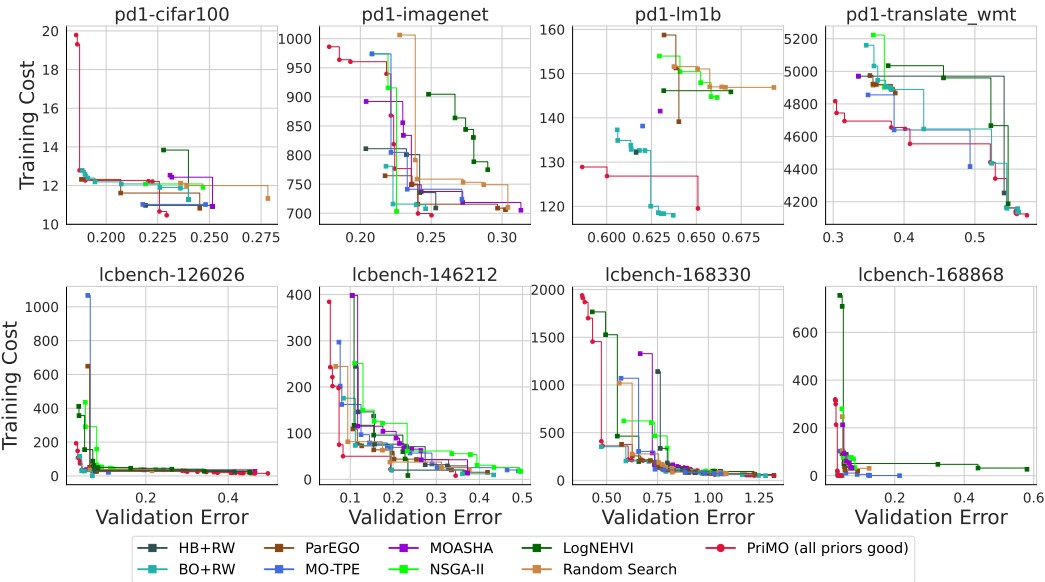

Figure 10: Shown here are the Pareto fronts obtained by `PriMO`, compared to other non-prior MO baselines under all good prior conditions.

## I   ADDITIONAL EXPERIMENTS AND ANALYSIS

In this section, we present detailed Hypervolume and Pareto plots across all 8 benchmarks under good and bad priors, comparing `PriMO` against both, non-prior and prior-based MO-adapted baselines.

### I.1   PERFORMANCE UNDER GOOD PRIORS

**Pareto Front analysis and comparison against non-prior baselines.**   For every baseline, we report the Pareto front aggregated across all seeds per benchmark in line with existing literature (Izquierdo et al., 2021; Schmucker et al., 2020; 2021), with the primary (validation error) objective on the x-axis and the training cost objective along the y-axis. The Pareto front plot in Figure 10 shows that, on average, `PriMO` and BO+RW locate the most non-dominated points compared to the other optimizers, however, `PriMO` clearly has the better Pareto Front coverage of the two across most benchmarks.

**Case-level comparison across all benchmarks**   In Table 15, we perform a case-level analysis across all benchmarks and compare the performance of `PriMO` and BO+RW based on the Pareto fronts aggregated across all seeds. We notice that, compared to standard BO+RW, `PriMO` achieves lower validation error across all benchmarks and lower training cost in most. We further notice that on 5 out of 8 benchmarks, `PriMO` clearly has **both** the lower error and model cost, further underlining `PriMO`'s better coverage of the Pareto front.

**Detailed Hypervolume comparison against prior-based baselines.**   In Figure 11, we present more detailed Hypervolume plots across all our 8 benchmarks, comparing `PriMO` against some prior-based baselines, adapted by us to the MO case. The plots demonstrate that, with the exception of the cifar-100 benchmark, `PriMO` is one of the two best optimizers across all benchmarks. This highlights `PriMO`'s ability to effectively utilize good priors despite the $\epsilon$-greedy non-prior based component of its BO. $\pi$BO+RWis marginally better than `PriMO` in some benchmarks, due to its much longer dependence on the priors. On the other hand, MO-Priorband seems to be quite ineffective in the utilization of good priors and is the worst performing HPO algorithm across most benchmarks.

Table 15: Case level comparison of performance of `PriMO` against BO+RW on 8 Deep Learning benchmarks based on aggregated Pareto fronts.

| Benchmark | `PriMO` Best Error | BO+RW Best Error | `PriMO` Min Cost | BO+RW Min Cost | Superior Algorithm |
|---|---|---|---|---|---|
| cifar-100 | **0.185** | 0.188 | **10.46** | 11.27 | `PriMO` |
| imagenet | **0.178** | 0.218 | **696.0** | 707.0 | `PriMO` |
| lm1b-transformer | **0.586** | 0.605 | 119.5 | **117.9** | - |
| translate-wmt-xformer | **0.303** | 0.347 | **4120** | 4140 | `PriMO` |
| LCBench-126026 | **0.0307** | 0.0390 | 14.97 | **1.43** | - |
| LCBench-146212 | **0.0509** | 0.0841 | **7.94** | 9.49 | `PriMO` |
| LCBench-168330 | **0.379** | 0.473 | 50.77 | **48.96** | - |
| LCBench-168868 | **0.0284** | 0.0353 | **0.00154** | **0.00154** | `PriMO` |

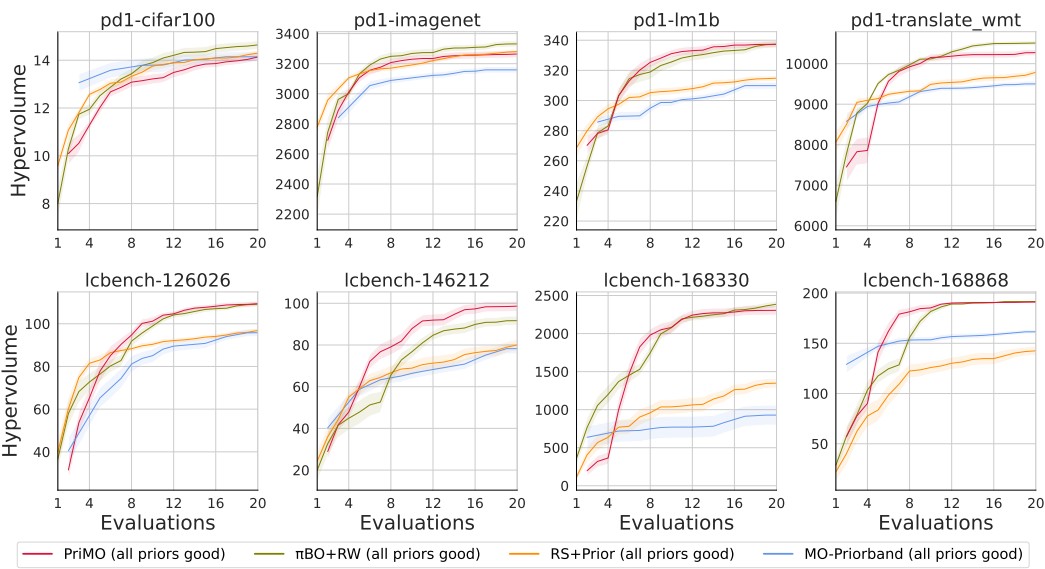

Figure 11: Comparing the average dominated HVI over 20 evaluations and across 25 seeds, between `PriMO` and prior-based baselines MO-Priorband, $\pi$BO+RW and RS + Prior, under all good prior conditions.

## I.2 ROBUSTNESS OF PRIMO UNDER BAD PRIOR CONDITIONS

Figure 12 shows `PriMO`'s remarkable ability to recover from bad priors with respect to the dominated Hypervolume, across all benchmarks. At the end of the optimization budget, `PriMO` achieves a competitive final performance, very close to BO+RW. Compared to prior-based baselines in Figure 13, `PriMO` is clearly shown to be the best performing optimizer across all benchmarks, with MO-Priorband - a close second. $\pi$BO+RW is unable to recover from bad priors and is the algorithm with the worst final performance on most benchmarks.

We attribute `PriMO`'s strong recovery under misleading priors to our design of the *decaying MO-prior-weighted acquisition*, influenced by two key parameters $-\beta$ and $\epsilon$. Unlike $\pi$BO+RW, which relies on the prior for much longer due to its slow decay schedule, `PriMO` is explicitly designed to reduce prior influence more aggressively. This design choice allows `PriMO` to recover more quickly when the priors are misleading, whereas $\pi$BO+RW's prolonged dependence on bad priors significantly hinders its performance, resulting in noticeably worse final performance compared to both `PriMO` and MO-Priorband. The aggressive $\beta$ setting ensures the prior's influence diminishes rapidly — an important property for practical DL scenarios where HPO is not expected to be run for long. Additionally, the parameter $\epsilon$ in the acquisition function controls how much the prior contributes while it is still active, thus encouraging exploration of the search space. Together, these two effects ensure that

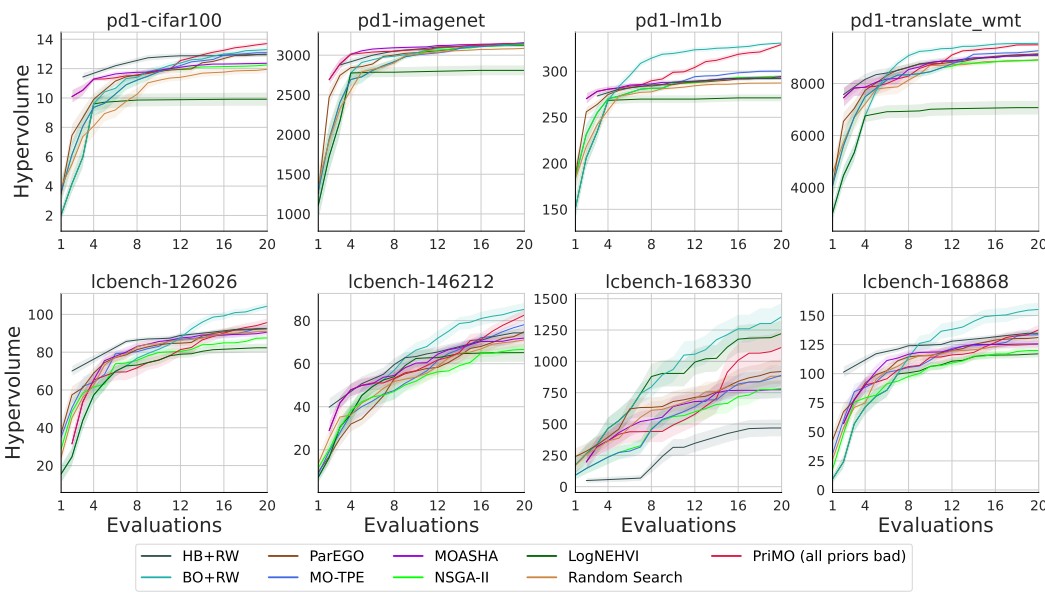

Figure 12: Mean dominated HV across 8 Deep Learning benchmarks, showcasing remarkable recovery of `PriMO` from bad priors, compared against some prominent non-prior baselines.

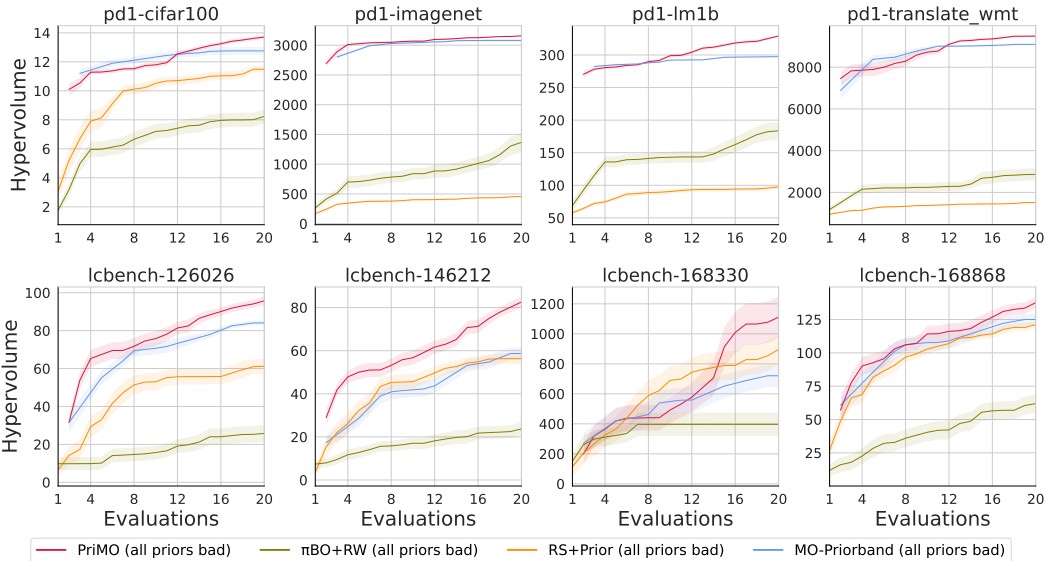

Figure 13: Dominated HV plot comparing `PriMO` against prior-based baselines that were adapted to MO, under bad priors.

`PriMO` does not become overly dependent on the prior and, under inaccurate priors, can still effectively explore and discover better hyperparameter configurations than its counterparts.

### I.3 A NOTE ON COMPUTE EFFICIENCY

`PriMO` stands out as being *extremely compute-efficient*, on average, achieving significant performance gains with minimal HPO evaluations, *i.e.*, with a low compute budget. We study this under overall prior conditions with respect to the dominated HV in Figure 14. Given that we set `PriMO`'s initial design size to 5, an asynchronous MF optimizer like MOASHA (in a continual setup) effectively requires only about 3.5 equivalent function evaluations, which

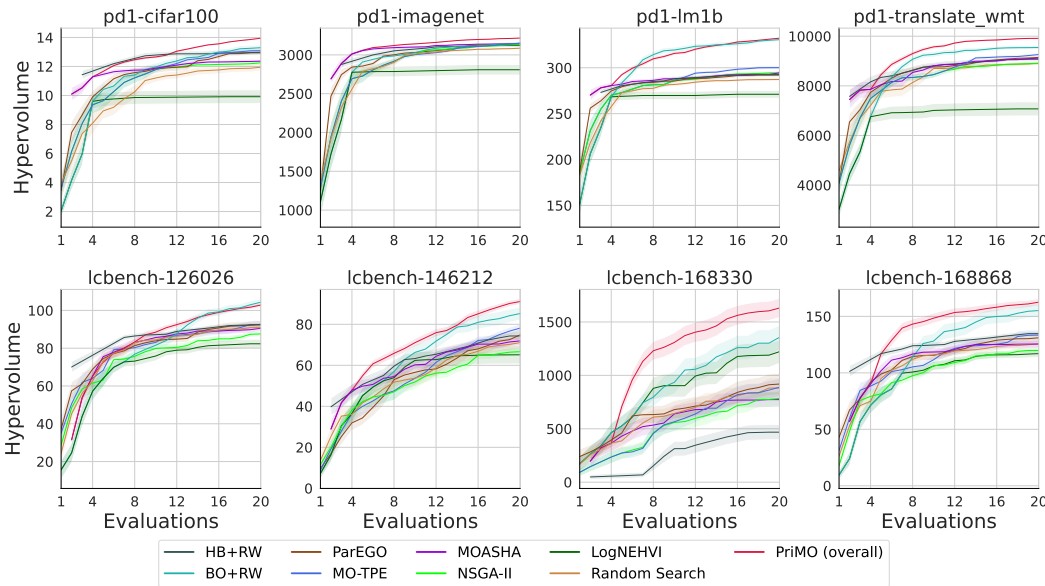

Figure 14: Comparing `PriMO` against some prominent non-prior MO baselines under the overall prior setting, with respect to the mean dominated HV across all 8 benchmarks.

on average results in 3 configurations sampled at $z_{max}$. Therefore, compared to other BO algorithms whose initial design size we set to the number of dimensions, `PriMO` effectively uses fewer max-fidelity configurations to fit the GP in the BO phase. Despite fewer samples, `PriMO` already achieves much better performance in the beginning compared to all BO-based baselines on most benchmarks, due to the use of its initial design strategy.

In summary, these findings support our claim that `PriMO` is a robust and general purpose multi-objective hyperparameter optimization algorithm designed for real-world DL workloads, fulfilling all the desiderata outlined in Table 1.

## J SIGNIFICANCE ANALYSIS

In this appendix, we perform statistical significance tests using Linear Mixed Effect Models (LMEMs) to verify the results obtained in our experiments. Our choice for using LMEMs is supported by Riezler & Hagmann (2022) who proposed LMEM-based significance testing for Natural Language Processing tasks. Further, Geburek et al. (2024) argued for the usage of LMEM-based significance analysis for HPO benchmarking.

### J.1 DATA PREPARATION AND SANITY CHECKS

To prepare the data for the significance analysis, we computed and used normalized Hypervolume regret scores, as the scale of HV can vary considerably across benchmarks. After aggregating the normalized HV regret values at each function evaluation, we conducted sanity checks to ensure statistical validity. We then performed a post-hoc analysis and used Critical Difference (CD) diagrams to compare the early and final performance of `PriMO` against all other algorithms.

**Seed independency check** We fitted two LMEMs:

$$\texttt{normalized\_hv\_regret} \sim \texttt{algorithm} \quad , \tag{11}$$

and,

$$\texttt{normalized\_hv\_regret} \sim \texttt{(0 + algorithm | seed)} \quad , \tag{12}$$

on the data and performed a Generalized Likelihood Ratio Test (GLRT) to verify that the seed is not a significant effect.

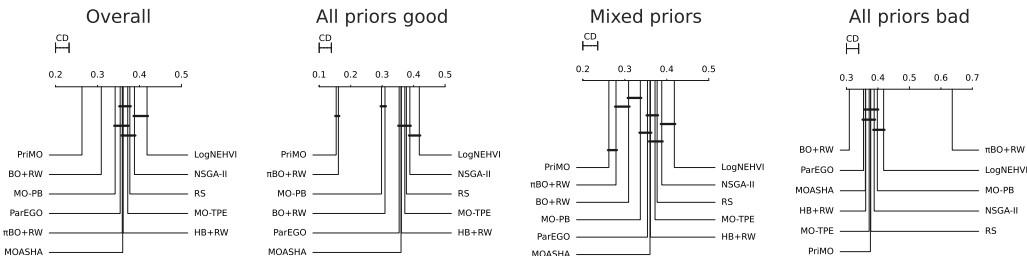

Figure 15: Critical Difference diagrams at **10 evaluations** comparing **early performance** of `PriMO` against the baselines – BO+RW, $\pi$BO+RW, MO-Priorband (MO-PB), MOASHA, HB+RW, MO-TPE, LogNEHVI, Random Search (RS), ParEGO and NSGA-II, under various prior conditions.

**Benchmark informativeness**  Using GLRT, we compared the likelihoods of the LMEMs:

$$\texttt{normalized\_hv\_regret} \sim \texttt{1} \quad , \tag{13}$$

and,

$$\texttt{normalized\_hv\_regret} \sim \texttt{algorithm} \quad , \tag{14}$$

which confirmed that our benchmarks are informative, as the second model (Equation 14) was shown to be significantly better. This further indicates that there are indeed significant differences between the performance of algorithms across all benchmarks, justifying the use of CD diagrams for comparison.

### J.2 CRITICAL DIFFERENCE DIAGRAMS

We perform pairwise Tukey HSD (Tukey, 1949) tests using LMEMs to obtain individual p-values for each comparison. Using this, we plot the CD diagrams.

Here, we consider the statistical differences in the early and final performance between `PriMO` and other algorithms. Figure 15 shows CD plots for 10 function evaluations, *i.e.*, halfway through our entire allocated budget. Figure 15 (all priors good) shows that `PriMO` is able to efficiently leverage good priors very early during the optimization, and significantly better than all baselines except $\pi$BO+RW. Under all bad priors, `PriMO`'s performance is not significantly worse than that of most the of best performing non-prior baselines, except BO+RW, but is significantly better than $\pi$BO+RW. However, averaging all prior conditions in Figure 15 (overall), we observe a significant difference between `PriMO` and all other optimizers. `PriMO` is shown to be the best ranked algorithm with significantly strong early performance.

In Figure 16, we show the CD diagrams for 20 function evaluations, *i.e.*, at the end of our optimization budget. As observed in our relative ranking plots before, there is negligible critical difference between `PriMO` and $\pi$BO+RW under all good priors, while both are significantly better than other algorithms. Figure 16 (all priors bad) verifies the final performance of `PriMO` under bad priors, highlighting a strong recovery, where, with the notable exception of BO+RW, `PriMO` is shown to be significantly better than all baselines. While not significantly better than BO+RW under mixed priors, `PriMO` is nevertheless the best ranked optimizer. overall, `PriMO` is clearly shown to be the algorithm with the highest rank, indicating the strongest final performance.

Thus, Figures 15 and 16 confirm our relative ranking plots and statistically verify `PriMO`'s state-of-the-art performance, proving that overall, `PriMO` is significantly better than all other algorithms used in our study.

## K  CODE REPOSITORY

The `Python` code for generating the priors and running the experiments presented in this paper is publicly available in `this` repository. This repository also contains the code used to

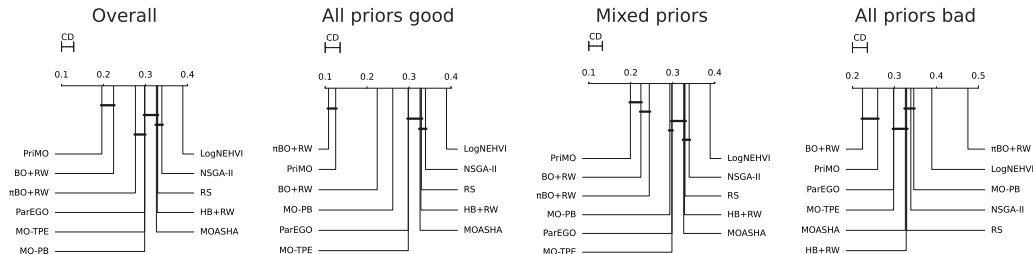

Figure 16: CD diagrams at **20 evaluations** comparing **final performance** of `PriMO` against other non-prior and prior-guided (adapted to MO) baselines, under various prior conditions.

generate all plots, along with a comprehensive `README.md` file that provides reproducibility guidelines, explains the output data structure, and outlines the steps required to run all baselines on the benchmarks used in this work. The priors over the objectives for the various benchmarks, and the raw results from the all optimization runs of `PriMO` used in this paper are also included.

## L  RESOURCES USED

We ran all the algorithms in this paper on inexpensive surrogate and synthetic benchmarks. To perform all our experiments, we only used a CPU compute cluster and 30 cores of Intel(R) Xeon(R) Gold 6242 CPU @ 2.80GHz. For runs up to 20 function evaluations, each seed of an HPO algorithm on a single benchmark took ∼0.025 CPU hours, or ∼0.75 core hours on average. While MF optimizers such as MOASHA and HB+RW completed in just a few seconds (∼0.15 core hours), model-based baselines such as BO+RW and $\pi$BO+RW required significantly longer on average – typically over 5 minutes (∼2.5 core hours).

For the experiments in Section 5, we ran 19 optimizers in total – 12 non-prior and 9 prior-based, including `PriMO` and all its design ablations, and the optimizers in the single-objective setting. Each prior-based multi-objective optimizer was evaluated under 4 different prior combinations, whereas 2 priors were used for the prior-based single-objective optimizers. Each run lasted 20 evaluations and we evaluated each optimizer on 8 benchmarks across 25 seeds. In total, this amounted to ∼208 CPU hours, or ∼6240 core hours to generate the results presented in Section 5.

## M  LICENSES

- NePS package: **Apache License, Version 2.0**
- `hpoglue`: **BSD 3-Clause License**
- `mf-prior-bench`: **Apache License, Version 2.0**
- Yahpo-Gym : **Apache License, Version 2.0**
- HyperBO  PD1: **Apache License, Version 2.0**
- Nevergrad: **MIT License**
- SMAC: **BSD 3-Clause License**
- Syne Tune (code for $\epsilon$-net): **Apache License, Version 2.0**
- `lmem-significance`: **MIT License**

