# OpenReview forum: "Multi-objective Hyperparameter Optimization in the Age of Deep Learning"
_ICLR.cc/2026/Conference — Submitted to ICLR 2026_

### Official Review · Reviewer_kZX4 · 2025-10-31

**Soundness:** 2
**Presentation:** 1
**Contribution:** 1
**Rating:** 2
**Confidence:** 3

**Summary:**

This paper proposes a multi-objective hyperparameter optimization method named PriMO that leverages expert prior knowledge. The approach integrates multi-objective expert priors into Bayesian optimization and utilizes cheap approximate surrogate models for initial design. Experimental results demonstrate that the method outperforms baseline approaches in both multi-objective and single-objective settings, while maintaining robustness to different prior strengths.

**Strengths:**

- The integration of expert prior knowledge with multi-objective optimization in hyperparameter tuning, as presented in this paper, represents a beneficial endeavor.
- The proposed method, PriMO, demonstrates superior performance in experiments, outperforming baseline methods in both multi-objective and single-objective settings.
- PriMO exhibits strong robustness to different prior strengths, and ablation studies confirm the effectiveness of each component of the framework.

**Weaknesses:**

- The description of the proposed method, PriMO, is relatively brief and lacks sufficient detail.
- The baseline methods used for comparison in the experiments are somewhat outdated (ranging from 2006 to 2021).
- There is a lack of case studies in real-world scenarios.

**Questions:**

*  Can the proposed method be applied to the fine-tuning or training of current popular LLMs?
*  How does the runtime efficiency of the proposed method compare to other approaches?
*  How is the expert knowledge introduced in this method defined, and can it be generalized to broader domains?

---

> ### Author Response · Authors · 2025-11-26
>
> Dear reviewer kZX4,
>
> We appreciate you assessing that our algorithm “demonstrates superior performance” in “both multi-objective and single-objective settings” with “strong robustness to different prior strengths”.
>
> In response to your comments and suggestions, we added additional baselines, performed a case study for LLMs, and expanded on the discussion and analysis of our algorithm. We reply to all your points in detail below and summarize our [major improvements in our reply to all reviewers](https://openreview.net/forum?id=1lLWZzikiT&noteId=NmIqbMN5WC)
>
> > The baseline methods used for comparison in the experiments are somewhat outdated (ranging from 2006 to 2021).
>
> We added two more recently published baselines [1, 2] in addition to the 9 baselines we already considered for a total of 11 (PriMO clearly remains state-of-the-art). We do want to point out that in the single-objective setting our original submission already considered algorithms published in 2022 and 2023 and the multi-objective baselines we constructed in addition to the ones present in the literature are modifications of these recently published algorithms.
>
> > There is a lack of case studies in real-world scenarios. [...] Can the proposed method be applied to the fine-tuning or training of current popular LLMs?
>
> We have added a case study on three benchmarking scenarios that cover the optimization of architectural parameters of a GPT2 style LLM (Section 5.5). PriMO is the strongest performing algorithm in this study.
>
> > The description of the proposed method, PriMO, is relatively brief and lacks sufficient detail.
>
> We have added more discussion on our algorithm (Section 4.4) and added a runtime analysis. Do you see any other concrete details as missing?
>
> > How does the runtime efficiency of the proposed method compare to other approaches?
>
> We added a discussion on the theoretical runtime behavior of PriMO and an empirical measurement of runtime efficiency (first paragraph of Section 4.4). In summary, PriMO’s asymptotic runtime corresponds to classic single-objective BO and empirically, PriMO’s runtime is negligible compared to Deep learning workloads.
>
> > How is the expert knowledge introduced in this method defined
>
> As we mention in our Limitations section, in line with related work, we consider Gaussian priors. The mean of the priors correspond to a configuration that is expected to perform well. In response to your question we have expanded on this and added forward references in Section 2.
>
> For the priors we use in our experiments, we adapt established protocols from the works on single-objective prior-based HPO published, e.g., in ICLR [3] and NeurIPS [4]. In our original submission in Section 5.1 (experimental protocol) under the paragraph titled “priors” we discuss this and refer to the full details on these protocols in Appendix E.
>
> We hope our reply could clear up your main concerns and if so, would appreciate a reassessment of our submission. We very much welcome any further feedback, suggestions, and discussions.
>
> References:
>
> [1]. K. Abe, Y. Wang, and S. Watanabe. Tree-structured parzen estimator can solve black-box combinatorial optimization more efficiently, 2025. URL https://arxiv.org/abs/2507.08053.
>
> [2]. S. Ament, S. Daulton, D. Eriksson, M. Balandat, and E. Bakshy. Unexpected Improvements to Expected Improvement for Bayesian Optimization. Advances in Neural Information Processing Systems, 36:20577–20612, December 2023. URL https://proceedings.neurips.cc/paper_files/paper/2023/hash/419f72cbd568ad62183f8132a3605a2a-Abstract-Conference.html.
>
> [3]. C. Hvarfner, D. Stoll, A. Souza, L. Nardi, M. Lindauer, and F. Hutter. πBO: Augmenting Acquisition Functions with User Beliefs for Bayesian Optimization. In The Tenth International Conference on Learning Representations (ICLR’22). ICLR, 2022.
>
> [4]. N. Mallik, C. Hvarfner, E. Bergman, D. Stoll, M. Janowski, M. Lindauer, L. Nardi, and F. Hutter. PriorBand: Practical hyperparameter optimization in the age of deep learning. In A. Oh, T. Naumann, A. Globerson, K. Saenko, M. Hardt, and S. Levine (eds.), Proceedings of the 37th International Conference on Advances in Neural Information Processing Systems (NeurIPS’23). Curran Associates, 2023

---

### Official Review · Reviewer_VfcQ · 2025-10-31

**Soundness:** 1
**Presentation:** 1
**Contribution:** 1
**Rating:** 2
**Confidence:** 4

**Summary:**

This paper addresses a specific issue in HPO, how to utilize prior knowledge in multi-objective HPO. It proposes a Bayesian optimization algorithm, PriMO, that integrates an initial design strategy and prior weights in BO steps. Experiments show that PriMO performs well in different cases, including all-priors-good, mixed-priors, and all-priors-bad.

**Strengths:**

- Utilizing prior knowledge in multi-objective HPO is a good, under-explored topic.

- Results exhibit good performance, whether the priors are good or bad.

**Weaknesses:**

- The title is too exaggerated in my eyes. HPO for deep learning faces numerous challenges, while the topic in this paper is only a very small one. Besides, it is not clear how the work addresses specific issues for deep learning.

- In practice, prior knowledge should be scarce and diverse. There is a lack of assumptions about the priors that this paper considers.

- The paper claims that priors can be good or bad. I wonder if it is a rigorous problem definition. How can you differentiate which ones are good or bad? If you cannot, how do you handle them differently?

- Figure 2 can not explicitly exhibit the motivation. First, it is not clear how to add prior on MOASHA or RS. Second, the advantages of adding prior in all-priors-good and MOASHA in all-priors-bad demonstrate the importance of how to differentiate good or bad priors, instead of the weakness of the naive method for adding priors.

- The main contribution comes from Eq. 4 and Algorithm 1. However, there is doubt that they are addressing issues related to the priors.

- Experiments are limited due to the diversity of the benchmark.

**Questions:**

See Weaknesses.

---

> ### Author Response · Authors · 2025-11-26
>
> Dear reviewer VfcQ,
>
> Thanks for acknowledging that we address a “good, under-explored topic” and recognizing our algorithms strength in performance and robustness.
>
> In response to your review we will change our title, have added discussion on several points to our paper, and performed additional experiments that extend our benchmarking further. We provide details below and list additional major improvements in [our reply to all reviewers](https://openreview.net/forum?id=1lLWZzikiT&noteId=NmIqbMN5WC
> ).
>
> Overall, we want to highlight that 4 of 6 weaknesses you list are connected to the concept of expert priors and not specific to the multi-objective setting we propose. Before addressing each comment individually, we want to point out that we follow established protocols and rely on concepts from the literature on single-objective HPO with expert priors published, e.g., at ICLR [1] and NeurIPS [2]. In response to your review, we now refer to respective details more clearly at key places and have added additional explanations and discussions on priors.
>
> We divide our rebuttal into two parts.
>
> ### Part 1
>
> > The title is too exaggerated in my eyes.
>
> We will change the title to “PriMO: Multi-objective Hyperparameter Optimization with Expert Priors”. However, we do want to point out that our old title mirrors the NeurIPS 2023 paper “PriorBand: Practical hyperparameter optimization in the age of deep learning” [2] that integrates expert priors for single-objective optimization.
>
> > The paper claims that priors can be good or bad. I wonder if it is a rigorous problem definition. How can you differentiate which ones are good or bad?
>
> To generate priors of different quality to use in our experiments, we follow established protocols from the works on single-objective prior-based HPO [1, 2]. In our original submission in Section 5.1 (experimental protocol) under the paragraph titled “priors” we discuss this and refer to the full details on these protocols in Appendix E.
>
> > “Figure 2 can not explicitly exhibit the motivation. First, it is not clear how to add prior on MOASHA or RS. Second, the advantages of adding prior in all-priors-good and MOASHA in all-priors-bad demonstrate the importance of how to differentiate good or bad priors, instead of the weakness of the naive method for adding priors.”
>
> We agree that how to add priors to MO approaches is a non-trivial question and argue this highlights our methodological contributions. The established protocols we follow allow for a clear distinction between good and bad priors. We now refer to these protocols already in the caption of Figure 2.
>
> In Section 3 and Figure 2 we show that the most straight-forward (“naive”) adaptation, i.e., directly sampling from the prior (or doing that 50% of the time) does not perform well, mirroring the argument for single-objective optimization presented in the Priorband paper [2]. We have added implementation details on these adaptations to Appendix F.

---

> > ### Author Response · Authors · 2025-11-26
> >
> > ### Part 2
> >
> > > In practice, prior knowledge should be scarce and diverse. There is a lack of assumptions about the priors that this paper considers.
> >
> > In our discussion on limitations (Section 7) we explicitly discuss the form of the priors we consider and in our experimental setup (Section 5.1) we discuss the priors we work with empirically. In response to your feedback, we now point to these discussions from Section 2 where we introduce multi-objective priors.
> >
> > We strongly disagree that prior knowledge is scarce and have added a discussion to Appendix D that mirrors the discussion in the literature. We see this as a settled discussion and also refer, e.g., to the [author-reviewer discussion of the Priorband paper](https://openreview.net/forum?id=uoiwugtpCH&noteId=DRNe4KcxL0) [2]. Prior knowledge could, for example, stem from the literature, prior tuning experience of a similar model, repeated experimentation, or from related problem settings. Furthermore, we would like to point out that resource objectives, which are very common in the multi-objective setting for DL, usually have intuitively very easy to define priors.
> >
> > > “The main contribution comes from Eq. 4 and Algorithm 1. However, there is doubt that they are addressing issues related to the priors.”
> >
> > Eq. 4, our acquisition function, directly incorporates the prior, addressing how to benefit from good priors while being robust to bad priors, e.g., by the integration of epsilon-BO and decay terms as discussed in Section 4.1. Our ablation study confirms that Eq. 4 is critical for the strong performance of PriMO (Algorithm 3), as all the worst ablations discard elements of Eq. 4.
> > In Algorithm 1, our initial design, we opted not to use the prior to increase overall robustness of PriMO (Algorithm 3) to the prior strength. Our ablation study confirms the improved performance across different prior strengths.
> >
> > > Experiments are limited due to the diversity of the benchmark.
> >
> > We have added an additional 3 benchmarks that cover the optimization of architectural parameters of a GPT2 style LLM, increasing our total benchmark count to 19 (11 for multi-objective optimization and 8 for the single-objective setting). For each of the two settings individually, our benchmark suite was already similar in diversity and size as related works published at ICLR and NeurIPS [1, 2].
> > We hope our reply could clear up your main concerns and if so, would appreciate a reassessment of our submission. We welcome any further feedback, suggestions, and discussions.
> >
> > References:
> >
> > [1]. C. Hvarfner, D. Stoll, A. Souza, L. Nardi, M. Lindauer, and F. Hutter. πBO: Augmenting Acquisition Functions with User Beliefs for Bayesian Optimization. In The Tenth International Conference on Learning Representations (ICLR’22). ICLR, 2022.
> >
> > [2]. N. Mallik, C. Hvarfner, E. Bergman, D. Stoll, M. Janowski, M. Lindauer, L. Nardi, and F. Hutter. PriorBand: Practical hyperparameter optimization in the age of deep learning. In A. Oh, T. Naumann, A. Globerson, K. Saenko, M. Hardt, and S. Levine (eds.), Proceedings of the 37th International Conference on Advances in Neural Information Processing Systems (NeurIPS’23). Curran Associates, 2023

---

> > > ### Comment · Reviewer_VfcQ · 2025-11-28
> > >
> > > Thanks to the authors for a detailed response. However, my concerns are still not fully addressed. I provide a summary as follows.
> > >
> > > Novelty issue:
> > >
> > > - This work obviously followed "PriorBand: Practical hyperparameter optimization in the age of deep learning". The primary ideas presented by Table 1, Figure 2 and "changing acquisition function weight", are very similar, which vastly diminishes the novelty.
> > >
> > > - The claimed key point of novelty is multi-objective. However, the only content that is related to multi-objective is Eq. 3 and MOASHA. Neither of which is novel.
> > >
> > > Reasonability issue:
> > > - Eq. 3 turns multiple objectives into a single objective through a randomly weighted sum of objectives following an old book (Yoon et al., 2009), which is not convincing.
> > >
> > > - MOASHA is used to provide an initial search space, which brings a high risk of missing optimal HP candidates.
> > >
> > > - The acquisition function weight, $\gamma$, is the primary part of the method. However, there is no explanation why priors from multiple objectives should be influenced by such an "uninformed" weight. There is no analysis of the distribution of multi-objective priors or the impacts of sampling methods. The experiment results lack sufficient theoretical support.

---

> ### Author Response · Authors · 2025-11-28
>
> Thanks for engaging in an active discussion and making further suggestions for improvements. In response, we improved our manuscript further as detailed below.
>
> > Novelty of the method
>
> The [reviewer guide](https://iclr.cc/Conferences/2026/ReviewerGuide) point 2.1 asks reviewers to assess the value and impact of papers differently depending on whether they "better address a known [...] problem" or "draw attention to a new [...] problem". Our main novelty is the introduction of a new problem: multi-objective HPO with expert priors (Section 2). Further, we show that directly adapting multi-objective algorithms to this new problem fails (Figure 2 and Section 3) and likewise directly adapting algorithms for single-objective HPO with expert priors (like PriorBand; Figure 3 and Section 5.3). Not only do we propose the first algorithm for this new problem setting, this algorithm is also simple and adapted to single-objective HPO with expert priors it also achieves state-of-the-art there (outperforming PriorBand).
>
> > Eq. 3 turns multiple objectives into a single objective through a randomly weighted sum of objectives following an old book (Yoon et al., 2009), which is not convincing.
>
> This old and simple scalarization outperforms all more recent algorithms except PriMO, including the ones we added in response to your review. Clearly random scalarization remains a relevant technique with convincing empirical results.
>
> > MOASHA is used to provide an initial search space, which brings a high risk of missing optimal HP candidates.
>
> No, MOASHA is used as an initial design (instead of Random Search) and our search space remains the same (see the first two sentences of Section 4.1 - An initial design to utilize cheap approximations). This is in contrast to how BO and multi-fidelity algorithms are usually combined, where the multi-fidelity aspect remains active throughout the optimization [1, 2].
>
> > there is no explanation why priors from multiple objectives should be influenced by such an "uninformed" weight.
>
> Gamma does not trade-off individual priors, instead it decays the contribution of all prior dependent parts of the acquisition function, such that, asympotically, PriMO's acquisition function (Eq. 4) tends to the original one (which could be any). We added this explanation to the method section and added a visualization (Figure 9) of how the contribution of the prior dependent parts diminishes with samples seen in Appendix C. Does this clear up the issues you had about Gamma?
>
> References:
>
> [1]. S. Falkner, A. Klein, and F. Hutter. BOHB: Robust and efficient Hyperparameter Optimization at scale. In J. Dy and A. Krause (eds.), Proceedings of the 35th International Conference on Machine Learning (ICML’18), volume 80, pp. 1437–1446. Proceedings of Machine Learning Research, 2018.
>
> [2]. L. Li, K. Jamieson, A. Rostamizadeh, E. Gonina, J. Ben-tzur, M. Hardt, B. Recht, and A. Talwalkar. A system for massively parallel hyperparameter tuning. In I. Dhillon, D. Papailiopoulos, and V. Sze (eds.), Proceedings of Machine Learning and Systems 2, volume 2, 2020.

---

### Official Review · Reviewer_PZA5 · 2025-11-01

**Soundness:** 4
**Presentation:** 3
**Contribution:** 3
**Rating:** 8
**Confidence:** 4

**Summary:**

This paper addresses the lack of HPO algorithms capable of incorporating multi-objective expert priors. While single-objective prior-informed optimization has received attention, extending this to multi-objective settings is both conceptually and technically nontrivial due to the need to reason over Pareto frontiers and conflicting objectives. The authors approach this issue starting from practical considerations in deep learning, looking at tradeoffs between accuracy, latency, cost, and fairness are common. The proposed PriMO framework provides a unified approach to integrate prior beliefs and cheap approximations while retaining robustness to misspecified priors.

**Strengths:**

1. The experimental section is definitive. The authors benchmark PriMO against a wide spectrum of baselines, ranging from classical multi-objective evolutionary algorithms to multi-fidelity optimizers (MOASHA, Hyperband) to Bayesian approaches. They also construct custom baselines (e.g., MOASHA+Prior, πBO+RW) to isolate the benefits of priors in the multi-objective context. PriMO consistently outperforms across eight deep learning benchmarks (image classification, translation, and language modeling) in both anytime and final performance metrics. It is rare in optimization to see such unilateral gains, which goes to show how underdeveloped the multi-objective HPO literature is and adds to the impact of this paper.

2. The authors keep practical considerations close to heart throughout the paper, which leads to very thorough investigation of relevant quantities like training cost and hypervolumes. This is very different from other approaches to multi-objective optimization, which tend to be either very theoretical and/or very complicated and engineered.

**Weaknesses:**

1. While the authors' acquisition function in Equation (4) works well empirically, the paper lacks a theoretical analysis of its properties. Ideal results would describe under what conditions we get convergence to the true Pareto frontier, or how the exploration parameter interacts with uncertainty estimation in BO. It is hard to know what the *secret sauce* of this choice is. I think this work would be stronger if there were some clear and simple example to have in mind that demonstrates the issue in multi-objective bilevel optimization which your approach solves/mitigates. I see that your algorithm looks reasonable and appears to do well, but in my opinion the most convincing results (and the ones that continue to hold at scale) are the ones with a clear "we unlock the ability to solve something other approaches completely fail at". Without knowing where to expect improvement to come from, it can make it very hard to refine and scale things.

2. It would be nice to have a more clear runtime analysis to understand exactly where the computation goes and how hyperparameters affect it. There is a lot of emphasis on wall-clock time improvement; the authors do well to demonstrate the improvement here, but in general it is good to have a theoretical asymptotic behavior to expect and aim to match/improve on.

**Questions:**

1. How do you envision practitioners specifying multi-objective priors? Would you consider incorporating structured or hierarchical priors for related tasks and subtasks?
2. I am naturally curious in all the theoretical properties: convergence to optimality, rates, robustness to noise, robustness to a bad prior, generalization to related problems, etc.. What do the authors expect based on the empirical anecdotal observations? What are the apparent strengths, weaknesses, and so on?

---

> ### Author Response · Authors · 2025-11-26
>
> Dear reviewer PZA5,
>
> Thanks for your detailed feedback and assessing the problem setting we introduce as “conceptually and technically nontrivial”. We also appreciate you recognizing our experiments as “definitive” and us keeping “practical considerations close to heart” all throughout.
>
> Below we address your suggestions and list the changes we made in response. In [our reply to all reviewers](https://openreview.net/forum?id=1lLWZzikiT&noteId=NmIqbMN5WC) we summarize all major improvements we made.
>
> > It would be nice to have a more clear runtime analysis to understand exactly where the computation goes
>
> We agree to the value of this and have added a brief runtime analysis to the new algorithm discussion subsection. As we use scalarization to transform the multi-objective problem to a single-objective one in each BO step, our runtime behaviour corresponds to classic BO.
>
> > In my opinion the most convincing results (and the ones that continue to hold at scale) are the ones with a clear ‘we unlock the ability to solve something other approaches completely fail at.”
>
> Other approaches in the literature do not consider the integration of expert priors at all. PriMO, thereby, unlocks the ability to do something that other approaches do not only fail at, but do not even attempt.
>
> > It is hard to know what the secret sauce of the algorithm is
>
> We answer this empirically in our ablation study (Section 5.6) that shows all components of PriMO are useful. The initial design provides good seed points for BO and is not affected by the presence of priors, which encourages a good initial exploration of the search space. The epsilon-BO leverages these good initial points and integrates prior knowledge to improve HPO performance, while simultaneously guarding against incorrect ones by decaying the prior strength quickly and interweaving non-prior-based acquisition function optimization.
>
> > How do you envision practitioners specifying multi-objective priors?
>
> We suggest adopting practices from the single-objective domain that build gaussian distributions around a believed-to-be-strong configuration, but to do so for every objective individually. In our limitations section, we mention that specifying a prior with respect to the pareto front could be more effective, however, how to do so remains a non-trivial question.
>
> We hope to have addressed your suggestions and would appreciate you championing our paper.

---

### Official Review · Reviewer_Bve8 · 2025-11-01

**Soundness:** 4
**Presentation:** 3
**Contribution:** 3
**Rating:** 6
**Confidence:** 4

**Summary:**

This paper presents PriMO, a prior-informed multi-objective hyperparameter optimization algorithm that extends Bayesian optimization with expert priors and a multi-fidelity initial design. The authors clearly motivate the gap that, while prior-based HPO methods exist for single objectives, no existing method supports multi-objective settings that are common in deep learning. PriMO introduces an ε-greedy acquisition strategy that balances prior guidance and exploration, and integrates a MOASHA-based warm-start to exploit cheap approximations. Extensive experiments across eight deep-learning benchmarks show strong anytime and final performance, outperforming state-of-the-art multi-objective and prior-based baselines.

**Strengths:**

The paper is very well written: definitions, algorithms, and experiments are presented cleanly and logically, making the work easy to follow. Multi-objective HPO is an important and practical topic for modern deep-learning workflows; addressing the lack of prior-aware solutions fills a real methodological gap. The proposed combination of prior-weighted acquisition with ε-greedy scheduling and a multi-fidelity initialization is conceptually coherent and empirically justified. The evaluation covers multiple domains, both surrogate and realistic, with clear ablation and robustness analyses that support the authors’ claims. Component-wise behaviors (e.g., prior strength, noise robustness, and early-stage acceleration) are analyzed in detail, giving the paper strong empirical credibility.

**Weaknesses:**

1. Beyond the Pareto-front visualizations, the paper could include more case-level examples or qualitative comparisons to help readers connect the optimization behavior with real task utility and model performance trade-offs.

2. In Algorithm 2 (the BO step), the parameter η is listed but seems unused—clarifying whether it affects fidelity scheduling or is inherited from the initialization stage would improve completeness.

3. A brief theoretical or intuitive discussion about how PriMO behaves when priors are highly correlated or partially redundant could further strengthen the understanding of its robustness.

**Questions:**

See weakness

---

> ### Author Response · Authors · 2025-11-26
>
> Dear reviewer Bve8,
>
> Thanks for assessing that we fill “a real methodological gap” and calling our paper “very well written” with “strong empirical credibility”. Below we address your suggestions and highlight the improvements we made in response to your feedback. In [our reply to all reviewers](https://openreview.net/forum?id=1lLWZzikiT&noteId=NmIqbMN5WC
> ) we list all major improvements.
>
> > Beyond the Pareto-front visualizations, the paper could include more case-level examples
>
> In Appendix I, in addition to the Pareto-front visualizations, we show hypervolume improvements on a case-level in our original submission. In response to your comment, we added a discussion and table to Appendix I with further case-level analysis.
>
> > A brief theoretical or intuitive discussion about how PriMO behaves when priors are highly correlated or partially redundant could further strengthen the understanding of its robustness.
>
> We added an intuitive discussion (Section 4.4) that addresses this and also investigate the effect of objective and prior correlations as part of our newly added case study on GPT2 like LLMs (Section 5.5). When priors are highly correlated, under the same kind of priors for both objectives (both good or both bad), the prior strength increases. PriMO’s effectiveness also increases when both priors are both helpful, thereby improving HPO performance. On the other hand, when both priors are bad, PriMO would initially struggle under the increased prior strength, but the epsilon-BO is particularly designed to be robust in such circumstances. However, under mixed prior conditions, the effects of the priors may cancel out or one prior may suppress the other depending on how strongly they are correlated.
>
> When priors are negatively correlated, the converse holds - we see performance gains under bad priors conditions on HW-GPT-Bench, and degraded performance under an average over mixed prior conditions.
>
> > In Algorithm 2 (the BO step), the parameter η is listed but seems unused
>
> Thanks, we now removed the parameter from the algorithm input.
>
> We hope we addressed your remaining concerns and if so, we would appreciate you considering raising your assessment scores.

---

### Author Response · Authors · 2025-11-26
**Overall response to initial reviews**

To all reviewers and chairs,

Overall, we appreciate the reviewers’ assessment that we introduce a “good, under-explored” (kZX4), “important and practical topic [that] fills a real methodological gap” (Bve8) which is “very different from other approaches to multi-objective optimization [and] conceptually and technically nontrivial” (PZA5).

We also appreciate that reviewers assess our experiments as “definitive” (PZA5) and “extensive” (Bve8) leading to “strong empirical credibility” (Bve8). In particular highlighting our “wide spectrum of baselines” (PZA5), our algorithms “superior performance [...] in both multi-objective and single-objective settings” (kZX4), its “strong robustness” (kZX4), i.e., “ its good performance, whether the priors are good or bad” (VfcQ”), and our ablation study that “confirm[s] the effectiveness of each component” (kZX4).

In response to the insightful and constructive reviews, we improved our paper in several ways. We list major changes below and discuss further improvements in our individual responses.
- We added a case study on GPT2 for language modeling (PriMO outperforms overall), increasing the diversity of our benchmarks and adding more experiments with LLMs.
- We added additional recently published baselines and PriMO clearly remains state-of-the-art in the multi-objective and single-objective setting.
- We added an empirical analysis of runtime efficiency (algorithmic runtime is negligible  compared to neural network training) and added a theoretical runtime analysis (PriMO behaves the same as Bayesian optimization), addressing questions on runtime efficiency.
- We added more discussion on priors and their sources, added a discussion of PriMO’s behaviour for correlated priors to the method section, and added an empirical analysis for our case-study, addressing the reviewers' questions on priors.

---

### Comment · Area_Chair_43XX · 2025-11-29

Dear Reviewers,

Authors’ kindly tried to address your concerns. If the responses address your concerns please acknowledge that. If not, please express remaining concerns. Thanks for your efforts!

Best, AC

---

### Meta-Review · Area_Chair_43XX · 2026-01-05

**Summary:**

The meta-review for "PriMO" (Submission 25300) centers on a fundamental disagreement between the practical utility of the tool and the novelty of its underlying components. While the paper addresses a relevant gap—incorporating expert priors into multi-objective HPO—the reviewers are polarized. Proponents (PZA5, Bve8) value the "definitive" empirical results. However, a significant block of reviewers (VfcQ, kZX4) argues that the method is a straightforward extension of PriorBand and lacks the conceptual or theoretical innovation required for a top-tier machine learning conference.

**Reviewer Concerns:**

Resolved: Benchmarking. The authors successfully expanded their benchmarks to include LLMs (GPT-2) and demonstrated that PriMO achieves state-of-the-art results compared to baselines like MOASHA and $\pi$BO.

Outstanding: Incremental Novelty. Reviewers kZX4 and VfcQ noted that the core of the algorithm (random scalarization and $\epsilon$-greedy acquisition weighting) borrows heavily from existing literature (Yoon et al., 2009; PriorBand). The rebuttal did not convince these reviewers that the transition to the multi-objective setting involved significant technical hurdles or novel insights beyond "combining existing parts."

Outstanding: Theoretical Justification. As noted by PZA5 and emphasized by the dissenting reviewers, there is no formal proof regarding the convergence to the Pareto frontier under the proposed acquisition function (Eq. 4). In a competitive field, the lack of $O(1/T)$ convergence rates or regret bounds for the multi-objective case is a notable weakness.

Outstanding: Prior Specification. Reviewer VfcQ argued that the paper lacks a rigorous definition of how multi-objective priors should be structured, especially when objectives are negatively correlated. The current "independent Gaussian" approach may be too simplistic for complex trade-offs.

**Reviewer Scores:**

No changes

---

### Decision · Program_Chairs · 2026-01-26

Reject